# Axiomatic Preference Modeling for Longform Question Answering

**Corby Rosset, Guoqing Zheng, Victor Dibia, Ahmed Awadallah, Paul Bennett**
Microsoft Research
corbyrosset, zheng, victordibia
hassanam, pauben@microsoft.com

## Abstract

The remarkable abilities of large language models (LLMs) like GPT-4 partially stem from post-training processes like Reinforcement Learning from Human Feedback (RLHF) involving human preferences encoded in a reward model. However, these reward models (RMs) often lack direct knowledge of why, or under what principles, the preferences annotations were made. In this study, we identify principles that guide RMs to better align with human preferences, and then develop an axiomatic framework to generate a rich variety of preference signals to uphold them. We use these axiomatic signals to train a model for scoring answers to longform questions. Our approach yields a **Preference Model** with only about 220M parameters that agrees with gold human-annotated preference labels more often than GPT-4. The contributions of this work include: training a standalone preference model that can score human- and LLM-generated answers on the same scale; developing an axiomatic framework for generating training data pairs tailored to certain principles; and showing that a small amount of axiomatic signals can help small models outperform GPT-4 in preference scoring. We intend to release our model.

## 1 Introduction

Recent advances in large language models (LLMs) has seen the introduction of diverse post-training strategies, including Reinforcement Learning from Human Feedback (RLHF) and Reinforcement Learning from AI Feedback (RLAIF). These techniques have helped bridge the "alignment gap" between the responses of raw pretrained language models and responses that resonate more closely with human preferences (Bai et al., 2022b; Ziegler et al., 2020). These techniques steer LLMs to prefer one response over another based on feedback signals from either human annotators, or from another LLM instructed to follow certain principles (Bahdanau et al., 2019; Kwon et al., 2023; Sun et al.,

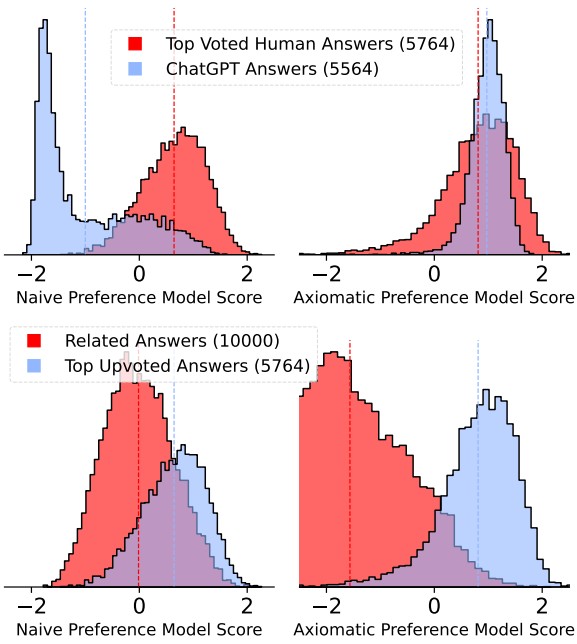

Figure 1: A naive preference model trained on upvotes *alone* is not aligned e.g., ChatGPT answers that are rated highly by humans are given low scores. An *axiomatic* preference model addresses this and other gaps.

2023). RLHF in particular involves the construction of a "reward model" (RM) which is trained to encode these human preferences and output a scalar score for any given response (Christiano et al., 2023; Stiennon et al., 2022; Beeching et al., 2023; Ouyang et al., 2022). Primarily, a RM-based approach to training LLMs separates *what* to learn from *how* to learn it (Leike et al., 2018).

The problem with most RMs used in RLHF post-training is that they are taught to regress a single scalar preference score annotated by humans without clear knowledge of why they made that decision or what principles they operated under. We term models trained in this fashion as *naive* preference-models. Furthermore, the underlying preference pairs used to train the RM do not come from diverse sources, often being sampled from the same

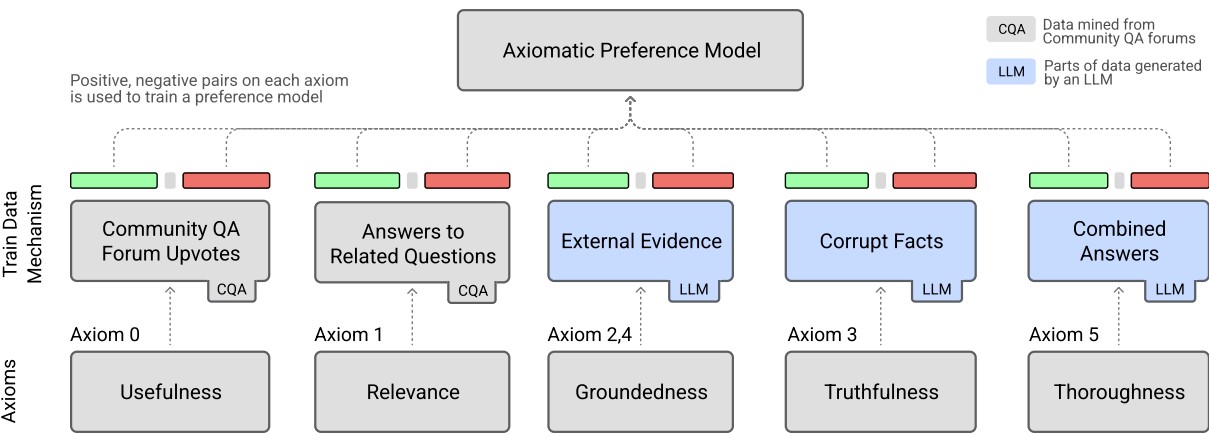

Figure 2: We propose five principled axioms to construct rich contrastive signals for training preference models

LLM they are trained on (Bai et al., 2022a; Nakano et al., 2022; Ouyang et al., 2022). It is also not clear that RMs can reliably score human-written and LLM-generated responses on the same scale, which is more challenging than previously anticipated due to vast differences such as style, as shown in Figure 1. Without clear signals of which principle informs the preference decision, and diverse sources of training examples upholding it, a RM may not be aligned with the expectations of human stakeholders.

For instance, studies have shown that RLHF-finetuned LLMs may fall short of key expectations – e.g. by failing to support claims with evidence, or making claims that sound convincing but are untrue – showing that there are still prevalent gaps in alignment for these scenarios (Liu et al., 2023; Zheng et al., 2023b; Menick et al., 2022).

In this work, we define principles (axioms) that humans desire in longform answers around the concepts of **usefulness**, **relevance**, **grounded-ness**, **truthfulness**, and **thoroughness** similar to (Thoppilan et al., 2022). A distinguishing feature of our study is that we then use these principles to construct candidate answer pairs "axiomatically" such that one answer is clearly preferred along a certain principle. Some of these axiomatic pairs are constructed from abundant sources of weak human preferences in the form of "upvotes" from Community-based Question Answering (CQA) sites like StackExchange [1]. In Figure 2 we illustrate how axiomatic pairs are generated for a single question. We define the principles in Appendix A, and describe how the axioms uphold those principles in Section 2.

Prior work used axioms to diagnose and correct failure modes in information retrieval systems (Fang et al., 2004, 2011; Rosset et al., 2019). Similarly, our axioms target known failure modes of modern LLMs, such as hallucinating incorrect statements that appear factual (Ji et al., 2023) or being distracted by irrelevant context (Shi et al., 2023). The axioms also enforce new capabilities, such as incorporating evidence, or addressing multiple perspectives. We believe our axiomatic framework provides richer, more targeted underlying preference pairs than, say, sampling from the same LLM with different temperatures.

Moreover, the RMs in existing studies are often not released nor the subject of close study compared to the LLMs they post-train. They can be quite costly, sometimes holding as many parameters as the LLMs they train. While there are many studies on RMs to address safety and toxicity issues (Bai et al., 2022a; Ganguli et al., 2022; Faal et al., 2022; Korbak et al., 2023; Ganguli et al., 2023), there are fewer on longform question answering (Nakano et al., 2022; Glaese et al., 2022).

Our approach is driven by the intuition that the act of identifying failure modes – or verifying an answer is free of them – is cognitively simpler (requiring fewer parameters) than the act of generating an answer, especially in the presence of authoritative evidence from a search engine. A separate, smaller RM also has many advantages: it is a controllable whitebox whose behavior is steerable, quantifiable, and decoupled from the LLMs it supervises; it allows for generalization to unseen examples without having to annotate them; and it is cheaper to run at scale.

The purpose of this study is to evaluate how well

---

[1] https://archive.org/details/stackexchange

| Principle | Axiom Description | Pair Construction |
|---|---|---|
| 0. Usefulness | Upvotes from CQA forums | If $A' >$ upvotes than $A''$
$\mathcal{PM}(Q, A') > \mathcal{PM}(Q, A'')$ |
| 1. Relevance | Answer, $A$, to $Q$ should be more relevant than answer $B$ to related question $Q'$, $Q' \in knn(Q)$ | A := Any Answer to $Q$
B := Answer to $Q'$
$\mathcal{PM}(Q, A) > \mathcal{PM}(Q, B)$ |
| 2. Grounded-ness | LLM Answer with context of relevant passages $P^+$ is better than without | C := LLM($Q$) "closed book"
D := LLM($P^+$, $Q$) "open book"
$\mathcal{PM}(Q, D) > \mathcal{PM}(Q, C)$ |
| 3. Truthfulness | LLM corrupts relevant answer $D$ yielding "wrong-but-believable answer" | E := LLM-Corrupt($D$, $Q$)
$\mathcal{PM}(Q, C) > \mathcal{PM}(Q, E)$
$\mathcal{PM}(Q, D) > \mathcal{PM}(Q, E)$ |
| 4. Relevant vs. Irrelevant Grounding | LLM answer with w/ relevant context $P^+$ is better than one w/ irrelevant context $P^-$ | F := LLM($P^-$, $Q$)
$\mathcal{PM}(Q, D) > \mathcal{PM}(Q, F)$ |
| 5. Thoroughness | Use an LLM to combine the top two user-upvoted answers, $A'$ and $A''$ | G := LLM-Combine($Q$, $A'$, $A''$)
$\mathcal{PM}(Q, G) > \mathcal{PM}(Q, A)$
$A \notin \{A', A''\}$ |

Table 1: Definitions of the axioms and how to construct training pairs from them based on our principles.

our proposed axiomatic RMs agree with human preferences. Hence, we refer to our model as a **Preference Model**, $\mathcal{PM}$ going forward. Note, using our preference models for LLM training (e.g. with RLHF) is outside of the scope of this paper. In Section 4 we demonstrate the capabilities of the our $\mathcal{PM}$ in several scenarios that require re-ranking candidate longform answers, including those written by humans and by LLMs.

The contributions of our work are threefold:

1. We develop an **axiomatic framework** to generate/augment training pairs that capture nuances in human preferences which may not be present in the existing data. These axioms can be tailored to enforce any well defined principle, meaning this framework is not limited to longform question answering.

2. We train standalone **preference models** $\mathcal{PM}$ (220M - 7B parameters) that can score both human- and LLM-generated answers on the same scale, normalizing out spurious signals such as length and style; our $\mathcal{PM}$ is better than training on human upvotes alone.

3. We show that training on the proper axiomatic signals boosts how well our $\mathcal{PM}$ agrees with both weak human upvotes and gold human annotators, even **exceeding the capabilities of GPT-4** – implying that GPT-4 may be overkill for preference scoring.

## 2 Axiomatic Preference Modeling

Learning a preference model for longform question answering can be formulated as a learning-to-rank problem (Cooper et al., 1992; Liu, 2009).

Given a question $q$ and a set of candidate answers $a_1, a_2, ..., a_n$, the goal of the preference model is to find a partial ordering of the answers by training on pairs that best align with real human preferences (Chen et al., 2013; Carterette et al., 2008). Existing neural architectures such as Transformer (Vaswani et al., 2017) are adept at solving learning-to-rank problems (Nogueira and Cho, 2020), and do even better under contrastive learning regimes (Xiong et al., 2020).

A preference model $\mathcal{PM}$ takes as input a question $q$, answer $a$, and outputs a scalar $\mathcal{PM}(q, a) \in \mathbb{R}$ a.k.a "preference score"; it has an optional input reserved for evidence passages $e$ denoted $\mathcal{PM}(q, e, a)$. We instantiate $\mathcal{PM}$ as a transformer-based cross-encoder (Wolf et al., 2019), $f$, whose input is a linearized sequence of tokens $x$ constructed from the concatenation of $q$ and $a$, denoted $x = q \odot a$. The output scalar is obtained from a linear regressor layer on the final transformer layer's CLS token. We further construct contrastive pairs of sequences such that the answer in one sequence $x^+ = q \odot a^+$ is preferred over a negative answer to the same question $x^- = q \odot a^-$. At training time, the sequences are fed into $f$ separately with the objective to score the positive example higher: $f(x^+) > f(x^-)$. We choose the margin loss to accomplish this goal:

$$\mathcal{L} = \max(0, \lambda - [f(x^+) - f(x^-)]) \quad (1)$$

where the margin, $\lambda$, between the positive and negative sequence in a pair can be fixed or computed. Importantly, while traditional learning-to-rank finds orderings based on relevance, we argue

that modern LLMs must go beyond that, which is why we introduce an expanded set of axioms including usefulness, thoroughness and groundedness.

## 2.1 Human Preference Signals

Learning to rank problems traditionally require a large set of candidates to re-rank. However, long-form answers are difficult to acquire. We turn to CQA forums such as Reddit and Stack Exchange specifically because questions there can receive *multiple* answers among which users can specify their preferences via "upvote" or "downvote" signals. Here we define axioms that produce training pairs either directly from CQA answers, or indirectly using LLMs; we list these in Table 1.

**Axiom 0 (Usefulness)** Critically, having multiple answers allows us construct preference pairs. We treat answers which have relatively higher upvotes as being more *useful* or *helpful*[2]. From the set of answers for a question $q$, we construct positive $a^+$ and negative $a^-$ training pairs such that $a^+$ has more upvotes than $a^-$ does.

Upvote signals are known to be noisy since users may upvote answers for various reasons, and may be influenced by position and presentation biases (Lee et al., 2016a). Answers can also gain popularity in a "rich get richer" fashion that may deviate from the intrinsic qualities of the answer itself (Lee et al., 2016b). However, upvotes generally aligns with our definition of usefulness (Fu and Oh, 2019).

**Axiom 1 (Relevance)** Answers in response to a question on a CQA site are more or less relevant, hence a model trained only on Axiom 0 would not have seen examples of off-topic answers. We imbue the training regimen with additional "hard negative" answers mined from related questions. We construct an KNN index of the ANCE embeddings for all questions in the Stack Exchange data dump (Xiong et al., 2020). For each question $q$, we retrieve $k$ nearest neighbor questions $\{q'\}_{i=0}^{k}$ (and all their constituent answers) from the same corpus such that the dot product of their vectors is below a chosen threshold $q \cdot q_i' < t_q$ to indicate $q_i'$ is related to $q$ while *not* being a paraphrase. This threshold $t_q$ is found manually. At training time, we randomly select $n$ negative answers across the union of answers to all $k$ related questions propor-

---
[2]helpfulness is part of the official answering guidelines of these CQA forums

tionally to their respective upvotes. By sampling negatives proportionally to their upvotes, we are able to specifically control for spurious signals such as length, style, presence of URLs, etc and force the model to inspect how the answer content interacts with the question.

## 2.2 LLM-generated Preference Signals

Axioms 0 and 1 leveraged upvotes to construct preference pairs from human-written answers. Here, we construct additional pairs generated by an LLM under various scenarios.

**Axiom 2 (Groundedness)** The Groundedness principle gives rise to a preference for an answer $a^+$ that incorporates and cites relevant evidence over one without access to such evidence, $a$. Hence negatives for a question $q$ come from an LLM (in our case, ChatGPT) in a "closed-book" style prompted with guidelines that mirror our principles. The "open-book" $a^+$ is generated from ChatGPT instructed to appropriately use evidence passages, $e$, placed in its context window, which were retrieved from the Bing API called with $q$ as the query. The prompt for this is shown in Figure 7 and examples in Figure 8.

**Axiom 3 (Truthfulness)** To combat hallucination of incorrect statements, we generate answers which intentionally corrupt factual claims in ways that are still believable. To do this, we take an open-book answer from Axiom 2 and instruct an LLM to deconstruct it into bullet-point claims, corrupt those claims individually, and then re-stitch the corrupted claims into a fluent answer, as shown in Figure 9; examples in Figures 10, 11. We found that open-book answers contain more factual statements, and hence have more to corrupt. We also found this prompting technique is the best way to automatically generate answers that are provably wrong without human annotation, otherwise, instruction-tuned LLMs would resist efforts to output false information. This corrupted answer should be worse than both an open-book and closed-book answer.

**Axiom 4 (Relevant vs. Irrelevant Grounding)** The sibling of Axiom 2, Axiom 4 targets the quality of grounding evidence because studies have shown that distracting context can be challenging for LLMs in longform QA scenarios (Krishna et al., 2021). Axiom 4 exploits relevance signals from a retrieval system to discern low quality passages $e^-$ from highly relevant ones $e^+$. To generate negative answers, we instruct an LLM to answer $q$ using

*only* information stated in $e^-$ and no other internal or external knowledge; see prompt in Figure 12. The positive $a^+$, on the other hand, is generated with access to $e^+$ in the same way as those in Axiom 2. We also construct additional training pairs among the evidence passages themselves to distill relevance signals directly into the $\mathcal{PM}$ as discussed in Appendix C.3.

While Axiom 2 used the Bing API for evidence, we need more fine-grained control of the retrieval scores to ensure $e^-$ is worse than $e^+$. We achieve this with the MS Marco dataset, which also has supervised relevance labels, by building a nearest neighbor index of the ANCE embeddings for all the documents (Xiong et al., 2020). For each $q$ in the MS MARCO training set, $e^+$ is collected from the top-$k$ documents plus those with a supervised relevance label of one; while $e^-$ are documents below a relevance threshold $t_{doc}$. The sets $e^+$ and $e^-$ do not overlap.

**Axiom 5 (Thoroughness)** The preference model should favor answers that better address the full scope of the question and all important perspectives. While this task is difficult to define, a simple yet effective approach is to assume that if two high quality answers $a'$ and $a''$ to $q$ come from two different authors, then their combination should be more thorough than either alone. We generate the positive $a^+$ from an LLM instructed to combine "the best of both worlds", $a^+ = \text{LLM-Combine}(q, a', a'')$. For training, $a^-$ are answers known to be worse than *both* $a'$ and $a''$, i.e. they have fewer upvotes. The prompt is shown in Figure 13 and examples in Figure 14. In practice, we select $a'$ and $a''$ to be the top two highest-upvoted answers on Stack Exchange, noting through extensive manual observations that users seldom upvote two answers with duplicate content very highly. We post-process this data to remove pairs where $a^+$ resembles naive concatenations its two constituents. For evaluation, we track $a^+$ vs $a'$ and $a''$ as in Table 3.

## 2.3 Connection to RLAIF & Constitutional AI

There is a strong connection between our Axiomatic framework described above and RLAIF. Firstly, the Axioms themselves build upon principles used to design LLMs like Lamda (Thoppilan et al., 2022). For instance, Claude's Constitution[3]

---

[3] https://www.anthropic.com/index/claudes-constitution

emphasized "helpfulness" and "honesty" which we operationalized into training pairs for Usefulness (Axiom 0) and Truthfulness (Axiom 3). Sparrow has a "stay on topic" Rule (Glaese et al., 2022) which we adapted as Relevance.

Secondly our Axiomatic framework is flexible enough to incorporate "AI feedback" from a much larger "teacher" model like GPT-4 by having it label/rank which axiomatic answer it prefers. However, we can go one step further and ask the teacher not only which it prefers, but *by how much* by scoring the answers. These fine-grained preference scores can learned by the $\mathcal{PM}$ via the $\lambda$ term in Equation 1, which governs the magnitude of separation between answers. Since answers we generate from LLMs lack upvote signals (and hence by default have a constant $\lambda$), this approach unifies learning from human- and AI-preference signals.

## 3 Experimental Methods

**Implementation Details** For all our experiments, the preference model is initialized from a T5Flan (Chung et al., 2022) base model. We train each model on a different combination of axiomatic pairs with a learning rate of 5e-6 warmed up linearly over 1k steps. We control for differences in training data size by mixing the data and training for exactly 16k steps – just under one epoch – to avoid any overfitting. We sample training examples uniformly at random according to the question (aka "posts") so that posts with many answers do not dominate. For each question, we group all pairs of its answers into the batch. The maximum sequence length of the concatenation of question, evidence, and answer tokens is 2048, with the question capped at 256.

**Data Collection** As a source of upvote data, we chose to mine and filter 905k posts from Stack Exchange across a variety of "substacks" covering topics ranging from biology to systems administration. There are about 3.4 answers per question on average, see Table 10. We filtered posts to those with at least two answers, one of which had positive upvotes, and at least one pair of answers where the higher had 30% more upvotes than the lower.

All questions used to seed LLM-generated axiomatic pairs were sampled from Stack Exchange above, except Axiom 4, which we constructed via MS Marco with evidence documents sourced from its corpus (Bajaj et al., 2018). Before training, we also confirmed that each type of answer pair con-

| | StackX | | r/ELI5 | | r/Science | | r/History | | MS Marco | | WebGPT |
|---|---|---|---|---|---|---|---|---|---|---|---|
| Avg. Ans per Q | 3.6 pos, 40 neg | | 4.6 pos, 43 neg | | 6.5 pos, 42 neg | | 5.3 pos, 47 neg | | 1.1 pos, 1k neg | | 1 pos, 1 neg |
| Metric | MRR | NDCG | MRR | NDCG | MRR | NDCG | MRR | NDCG | MRR | NDCG | Accuracy |
| length(Ans) | 15.0 | 35.4 | 6.2 | 27.6 | 7.7 | 30.1 | 15.0 | 37.1 | n/a | n/a | 56.7 |
| OpenAsst-RM 6.7B | 25.0 | 44.6 | 12.7 | 34.7 | 15.4 | 38.1 | 24.4 | 46.1 | 4.0 | 17.3 | **76.5** |
| StackLlama RM 7B | 26.8 | 45.1 | 8.3 | 30.6 | 10.3 | 33.3 | 9.8 | 33.1 | 3.4 | 16.1 | 56.1 |
| GPT-4 (listwise) | 45.5 | 62.1 | 39.6 | 59.9 | 35.1 | 56.4 | 37.8 | 60.4 | n/a | n/a | n/a |
| $\mathcal{PM}_0$ T5-base | 31.2 | 48.6 | 11.1 | 32.6 | 14.8 | 37.0 | 24.0 | 44.5 | 3.9 | 16.9 | 51.1 |
| $\mathcal{PM}_{0\text{-}1}$ T5-base | 64.3 | 78.8 | 54.5 | 75.2 | 53.2 | 75.4 | 63.1 | 84.3 | 16.1 | 30.6 | 55.7 |
| $\mathcal{PM}_{0\text{-}2}$ T5-base | 65.5 | 79.8 | 55.1 | 76.3 | 51.9 | 74.6 | 61.4 | 83.1 | 9.7 | 25.6 | 57.6 |
| $\mathcal{PM}_{0\text{-}3}$ T5-base | 65.3 | 79.5 | 55.0 | 76.0 | 51.4 | 73.9 | 61.1 | 82.8 | 9.4 | 23.7 | 55.4 |
| $\mathcal{PM}_{0\text{-}4}$ T5-base | 65.8 | 80.0 | 54.0 | 75.2 | 51.1 | 74.0 | 61.2 | 83.0 | 25.0 | 39.3 | 58.6 |
| $\mathcal{PM}_{0\text{-}5}$ T5-base | 64.6 | 79.2 | 53.6 | 75.0 | 51.6 | 74.3 | 61.7 | 83.3 | 23.1 | 37.4 | 58.1 |
| $\mathcal{PM}_{0\text{-}5}$ T5-large | 66.4 | 80.8 | 55.9 | 77.0 | 55.4 | 77.2 | 64.0 | 85.4 | 24.3 | 38.9 | 59.1 |
| $\mathcal{PM}_{0\text{-}5}$ Llama-7b | **74.9** | **86.7** | **65.5** | **85.6** | 60.5 | **82.5** | 69.6 | **89.5** | **37.5** | **50.1** | 59.9 |
| $\mathcal{PM}_{0\text{-}5} + \lambda$ Llama-7b | **74.9** | **86.7** | 65.3 | 85.4 | **60.8** | 82.4 | **69.7** | **89.5** | 31.5 | 45.1 | 61.3 |

Table 2: We evaluate $\mathcal{PM}$ on answer ranking tasks, trained under various combinations of axioms. Ranking is performed in the presence of "hard negatives" from semantically related questions (or BM25, for MS Marco). We compare against open-source reward models: Stack-LLama and OpenAssistant, both of which have 7B parameters. Our $\mathcal{PM}$ were not trained on WebGPT data (but OA-RM was); StackLLama was trained on Stack Exchange.

structed by the Axioms was indeed preferred by humans, as shown in Table 6. Any pair whose positive was preferred less than 70% of the time was removed from training. We discuss more in Section 4.1. For mining related questions in Axiom 1, we set $k = 10$ which leads to about 40 hard negative answers on average per post. Table 7 shows the sizes of our datasets, Appendix C explains more.

**Choosing the Margin** We found computing a margin of $\log_{10}(\text{votes}(a^+)/\text{votes}(a^-))$ to work best, congruous with (Askell et al., 2021). For LLM-generated answer pairs (where upvotes do not exist), the margin was a fixed constant of 0.25. The only exception is for $\mathcal{PM} + \lambda$, where GPT-4 was first asked to "critique-then-score" each answer on a scale of 1-100 in a listwise fashion, and then the margin was computed after filtering for pairs where the score difference was at least 5.

**Existing Open-source Baselines** We also evaluate against two 7B-parameter reward models publicly available: one that was used to train Huggingface's StackLLama model[4] and another used to train OpenAssistant[5] from Laion AI.

### 3.1 Evaluation

We evaluate our $\mathcal{PM}$ using the following datasets and quality metrics:

**Held-out Stack Exchange** set of 5.5k posts, with all their respective human-written answers

[4]llama-7b-stack-exchange-RM-peft-adapter-merged
[5]oasst-rm-2-pythia-6.9b-epoch-1

and LLM-generated answer pairs from Axioms 1, 2, 3 and 5. We evaluate quality in a ranking setting by ordering human-written answers along the $\mathcal{PM}$ scores, and compute the MRR of the top-upvoted answer as well as NDCG (Järvelin and Kekäläinen, 2000, 2002). We also evaluate accuracy on held-out axiomatic pairs for Axioms 2, 4, and 5.

**ELI5 Test** set of about 20k questions across the r/ELI5, r/Science, and r/History subreddits (Fan et al., 2019). This data has a similar format to Stack Exchange since there are multiple user-written answers to a posted question which other users can upvote. Hence, we evaluate MRR and NDCG as in Stack Exchange above. For increased difficulty of answer ranking, both ELI5 and Stack Exchange held-out data contain hard-negative answers to related questions *à la* Axiom 1, where all negatives are set to have a relevance gain of 0.

**WebGPT Comparisons** dataset of about 19.5k questions, each with a pair of retrieval-augmented answers collected from a LLM-based web browsing assistant named WebGPT (Nakano et al., 2022). Each pair of answers has human preference annotations, on which we compute accuracy of whether the $\mathcal{PM}$ gives a higher score to the preferred answer; we also confirm statistical significance of our results by showing the p-value from a student's T-test. We evaluate only on the 17,622 answer pairs which had a "clear" preference. The preferred answers had about $137 \pm 41$ words compared to $127 \pm 46$ for the negatives.

| | Ax 2: Open- vs Closed Book | | | Ax 4: Rel.- vs. Irrel. Context | | | Ax 5: Combine Top 2 | |
| --- | --- | --- | --- | --- | --- | --- | --- | --- |
| | Pos >Neg | with Evidence $e^+$ | | Pos >Neg | with Evidence $e^+$ | | Comb >1st | Comb >2nd |
| | Acc (%) | Acc (%) | $\Delta$ Pos | Acc (%) | Acc (%) | $\Delta$ Neg | Acc (%) | Acc (%) |
| $\mathcal{PM}_0$ T5-base | 70.0 | 64.0 | -0.18 | 30.9 | 19.4 | -0.06 | 25.7 | 34.9 |
| $\mathcal{PM}_{0-1}$ T5-base | 77.7 | 53.9 | -0.55 | 52.8 | 20.2 | -0.29 | 47.0 | 57.7 |
| $\mathcal{PM}_{0-2}$ T5-base | 76.4 | 69.5 | -0.058 | 82.3 | 54.5 | +0.27 | 66.3 | 80.3 |
| $\mathcal{PM}_{0-3}$ T5-base | 71.3 | 22.8 | -0.38 | 76.0 | 87.7 | -0.53 | 58.2 | 73.8 |
| $\mathcal{PM}_{0-4}$ T5-base | 55.1 | 73.7 | +0.059 | 91.4 | 98.4 | -0.27 | 59.7 | 75.8 |
| $\mathcal{PM}_{0-5}$ T5-base | 53.4 | 79.0 | +0.089 | 92.8 | 98.1 | -0.094 | 97.4 | 98.6 |
| $\mathcal{PM}_{0-5}$ Llama-7b | 74.3 | 72.1 | -0.01 | 90.3 | 97.1 | +0.01 | 99.0 | 99.2 |
| $\mathcal{PM}_{0-5} + \lambda$ Llama-7b | 81.3 | 73.3 | -0.09 | 89.6 | 94.8 | -0.094 | 59.0 | 78.4 |

Table 3: Evaluation on held-out pairs for axioms 2, 4 and 5. We evaluate answers with and without the evidence used to construct them, where positives are supposed to have higher scores in presence of their grounding evidence.

**MS Marco** passage ranking dev set has 6.9k questions, each with 1k BM25-negative passages and around one supervisedly labeled relevant passage (Bajaj et al., 2018). We use our $\mathcal{PM}$ to rerank all ~1k passages and compute MRR and NDCG. Note, held out data for Axiom 4 used the passages to augment LLM-generated answers to the dev questions; here we rerank the passages themselves.

**"Research-Analysis Questions"** of 500 difficult, hand-curated questions that go beyond factoid questions to elicit more intense reasoning and longer form answers which require multiple perspectives and evidence. They have no one right answer. We describe this dataset more in Appendix D and show multiple examples in Figure 9. We generate multiple candidate answers, pair them, and get gold human preferences among the pairs. We then compute agreement between $\mathcal{PM}$ and the gold preferences as described in Section 4.3.

Data from Stack Exchange and MS Marco were used for training the $\mathcal{PM}$ and are considered "in-domain". We do not train on data from Reddit ELI5, WebGPT or Research Analysis Questions.

## 4 Results

Throughout these results, we compare preference models trained on various combinations of the axiomatic data, e.g. "$\mathcal{PM}_{0-2}$" denotes training with data pairs from Axioms 1 and 2 *added* to the original pairs from Axiom 0.

### 4.1 Evaluating Axiomatic Data Construction

Our first goal is to compare human and LLM-written answers on the same scale. Qualitatively, we expect a good $\mathcal{PM}$ to score answers to related questions (Axiom 1) on the lowest end of that scale (since they don't even address the question at hand), followed by human-written answers with relatively

low or negative upvotes. On the other hand, most answers generated by ChatGPT (a capable LLM) should score highly, similar to the highest-upvoted human-written answers.

Figure 1 shows that $\mathcal{PM}_0$ (a naive model trained only on upvotes) falls short of these expectations, which we believe is due to stylistic differences in LLM-generated answers, noise in the upvote signals, and lack of meaningfully irrelevant answers naturally occurring in Stack Exchange posts. A more detailed qualitative comparison in Figure 4 shows that $\mathcal{PM}_{0-1}$ is good but not sufficient and that $\mathcal{PM}_{0-2}$ is the "minimum" amount of axiomatic signals needed to correct these issues.

Table 6 shows our efforts to verify that each type of axiomatically constructed training pair is indeed aligned with human preferences, and if not, it is disqualified from the training set. The annotators indicated their preference on a 6-point scale ("Strongly Prefer A", "Moderately Prefer A", "Slightly", etc) without the option for a tie. These results also confirmed that often times the ChatGPT-generated answers were preferred to the top-upvoted human answer (57% to 43%).

Our conclusion is that a combination of axiomatic training signals is needed for a $\mathcal{PM}$ to abide by the principles and score human- and LLM-written answers on the same scale, without overfitting to spurious signals. Put another way, the axioms *regularize* noisy user upvote signals.

### 4.2 $\mathcal{PM}$ for Answer Ranking

In Table 2 we evaluate $\mathcal{PM}$ in answer ranking settings, showing the average number of positive and negative answers per task. As a baseline, we also have GPT-4 rank these answers "listwise" (meaning in a single completion call, GPT-4 must output the new order of the answer ids given a context

| Answer Pair (A vs. B) | Prefer A >B (%) | | | Agreement w/ 3-Way Human Annotators (%) | | | | | |
|---|---|---|---|---|---|---|---|---|---|
| | Human | GPT-4 (tie) | $\mathcal{PM}_{0\text{-}5}$ | GPT-4 (tie) | $\mathcal{PM}_{0\text{-}5}$ | 0-4 | 0-2 | 0-1 | 0 |
| GPT-4 vs ChatGPT | 94.0 | 94.0 (4.1) | 83.2 | 92.7 (2.0) | **82.0** | 80.4 | 66.4 | 16.0 | 28.0 |
| GPT-4 vs "GPT-4 fixing Vicuna13B" | 79.6 | 51.5 (26.2) | 74.1 | 72.8 (4.1) | **73.2** | 71.6 | 60.4 | 36.4 | 44.8 |
| GPT-4 vs "GPT-4 Plan & Search" | 74.4 | 68.2 (19.6) | 75.5 | 69.9 (6.9) | 66.4 | **70.4** | 57.6 | 37.6 | 44.0 |
| "GPT-4 fix V" vs "GPT-4 P&S" | 45.2* | 48.0 (22.0) | 44.1 | 58.9 (11.0) | **60.8** | 55.6 | 59.2 | 40.4 | 43.6 |
| "GPT-4 fix V" vs "Vicuna13B P&S" | 76.0 | 52.0 (20.5) | 58.7 | 64.6 (16.3) | 64.4 | **67.6** | 52.4 | 33.2 | 34.0 |
| "GPT-4 P&S" vs "Vicuna13B P&S" | 82.4 | 41.2 (24.7) | 65.2 | 47.6 (20.3) | **63.2** | 50.0 | 43.2 | 36.0 | 38.4 |
| "Vicuna13B P&S" vs ChatGPT | 52.8* | 76.0 (10.3) | 43.0 | 65.5 (1.6) | 60.0 | **63.2** | 55.6 | 42.0 | 43.6 |
| "Vicuna13B P&S" vs Vicuna13B | 59.5 | 61.2 (11.5) | 60.5 | 67.3 (4.6) | 65.4 | **66.1** | 59.3 | 37.4 | 38.0 |
| Vicuna13B vs ChatGPT | 31.2 | 55.8 (19.2) | 35.3 | 47.2 (17.5) | 67.2 | **68.4** | 51.6 | 26.0 | 30.0 |
| Overall Agreement: | | | | 65.4 (8.9) | **66.8** | 65.9 | 56.5 | 34.2 | 38.3 |

Table 4: Human judges are asked to annotate gold preferences on pairs of answers to a hand-crafted set of 500 difficult "Research Analysis Questions". We compare how well various $\mathcal{PM}$ agree with their preference decision.

containing all the answers). Our results show that despite the advantage GPT-4 has in seeing all the answers at once, our $\mathcal{PM}$ can still align with noisy human preference signals better than GPT-4 with only about 220M parameters. Notably, $\mathcal{PM}_0$ falls short for this task, due to its inability to distinguish the top-upvoted answers from the hard negatives. For the MS Marco passage reranking task we note that BM25 achieves a MRR of 18.4, which is exceeded only after incorporating Axiom 4's data.

It is also surprising that existing reward models like OpenAssistant-RM and StackLlama fall short of expectations on these re-ranking benchmarks, especially since StackLLama was trained on Stack Exchange as well. It appears that for preference modeling, the quality of training signals is more important than the size of the model.

In Table 3 we evaluate the $\mathcal{PM}$ on held-out pairs of answers constructed by Axioms 2, 4 and 5. If a $\mathcal{PM}$ is not trained on one of the axioms in this table, that axiom is considered a zero-shot evaluation. A key performance indicator of a well-grounded $\mathcal{PM}$ is giving higher scores to answers $a^+$ that properly cite supporting evidence $e^+$ against closed-book answers $a$ (Axiom 2), or against those answers $a^-$ that cited irrelevant evidence $e^-$ (Axiom 4). When given access to $e^+$ in column two of Table 3, the $\Delta$ between $\mathcal{PM}$ $(q, e^+, a^+)$ and $\mathcal{PM}$ $(q, a^+)$ should be positive, indicating the $\mathcal{PM}$ is more confident that $a^+$ is superior to $a$, resulting in higher accuracy.

Similarly for Axiom 4, giving the $\mathcal{PM}$ $(q, e^+, a^-)$ access to $e^+$ makes it more apparent that $a^-$ is omitting, or even at odds with, the relevant information in $e^+$. In other words, higher accuracy with access to $e^+$ means it is easier to detect $a^+$ is better than $a^-$ than without access.

The last two columns of Table 3 show that, as intended, the positive answer from Axiom 5 is better than the top two upvoted answers it LLM combined in the first place; and additionally, it is found to be more superior to the second highest upvoted answer than the first.

### 4.3 $\mathcal{PM}$ Agreement with Gold Human Preferences

We generate a set of answers to hard "Research Analysis Questions" from different models like ChatGPT, GPT-4 (OpenAI, 2023), and Vicuna13B (Chiang et al., 2023). We also prompt them under different scenarios such as using tools like the Bing API to iteratively "Plan & Search" before synthesizing a final answer (Schick et al., 2023), or using GPT-4 to fix Vicuna13B's "Plan & Search" attempt in a feedback loop ("GPT-4 fix Vicuna") (Madaan et al., 2023; Welleck et al., 2022). We intend these scenarios to reflect real-world use cases of LLMs. We then select *pairs* of answers to send for gold human preference labeling, which leads to better calibration than scoring them individually (Carterette et al., 2008; Ziegler et al., 2020). Per answer pair, at least three annotators provide a 6-point preference score with Fleiss kappa $\kappa = 0.42$ indicating good inter-rater agreement. More details are in Appendix B.1.

We then evaluate in Table 4 how well our $\mathcal{PM}$ agrees with the human gold preferences. We define agreement as: if the majority of the annotators preferred answer A over B, did the $\mathcal{PM}$ give a higher score to A, and vice versa. An * means not statistically significant. As a baseline, GPT-4 was again prompted to score answers "listwise" with critique-then-score technique (Appendix B.3 and Figure 5) similar to (Wang et al., 2023). Hence, GPT-4 had the advantage of access to more answers for bet-

ter preference calibration, while the $\mathcal{PM}$ was at a disadvantage because it only scores an answer "pointwise" at test time. We record when GPT-4 gave a tie to an answer pair. Despite GPT-4's advantage, our **220M parameter $\mathcal{PM}$ 0-5 has higher agreement with gold human preferences**. Table 4 also shows that a mixture of multiple axioms is needed to exceed 50% agreement, which is the random choice baseline.

### 4.4 Constant vs. Variable Margins

Lastly, in Figure 3 we show the qualitative differences between a $\mathcal{PM}$ 0-5 llama2-7b trained with a constant margin for all LLM-generated axiomatic training pairs, versus one with a variable margin derived from GPT-4 preference scores. While the re-ranking evaluations in Table 2 for these two models do not show much variation, this histogram reveals that even large preference models which see both human- and LLM-generated answers can be vulnerable to overfitting on the style/length of LLM answers. We believe that fine-grained AI-feedback scores from a model like GPT-4 can help defend against this.

## 5 Related Work

Early works on scoring LLM outputs like LaMDA and BlenderBot3 collect scores of single input-output pairs rather than preference scores between pairs of candidate outputs (Shuster et al., 2022; Thoppilan et al., 2022). More recent reward models (RMs) fall into two camps. The first is training separate regressor models like those used in RLHF, which are often a single reward model to encode a one dimension of human preferences (Böhm et al., 2019; Ziegler et al., 2020; Bahdanau et al., 2019; Ouyang et al., 2022; Korbak et al., 2023) or many dimensions (Bai et al., 2022a; Ramamurthy et al., 2022). The second camp uses LLMs instructed to give feedback based on principles or a "constitution" (Bai et al., 2022b; Kwon et al., 2023), with the drawback of being costly to query.

Other approaches seek more fine-grained reward signals or multiple reward models, such as collecting relevance, correctness, and completeness signals on both sentence- and response-levels using separate reward models (Wu et al., 2023). Sparrow collects "targeted judgements" from human annotators to better characterize which of 23 rules a LLM violated (mostly around toxicity and safety), and then train multiple targeted classifiers (Glaese et al.,

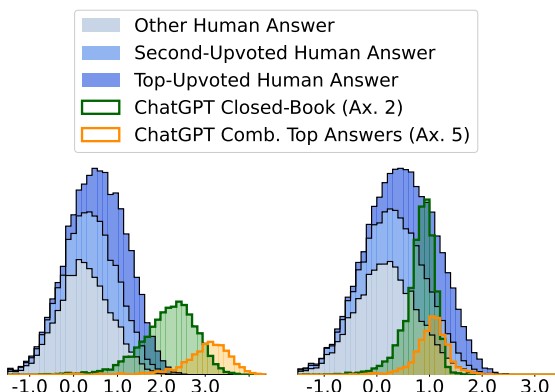

Figure 3: Distribution of $\mathcal{PM}$ 0-5 scores on both human- and ChatGPT-generated answers to our Stack Exchange dev set. The (left) was trained with a constant margin whereas the (right) $\mathcal{PM}$ 0-5 $+ \lambda$ was trained with GPT-4-annotated preference margins *per training pair*.

2022). The coexistence of rule-based and trained reward functions is also explored in (Ramamurthy et al., 2022).

Process supervision has emerged as a promising direction to provide feedback at each step of a complex multi-step task (Lightman et al., 2023).

Retrieval augmentation has been shown in several studies to mitigate hallucination of incorrect statements in LLMs, by either finetuning LLMs with grounding documents (Lewis et al., 2020), or inserting them to the context windows without fine-tuning LLMs (Ram et al., 2023). Other methods infuse retrieved knowledge in the decoding stage for knowledge-intense question-answering tasks (Liu et al., 2022).

## 6 Conclusions

We show that augmenting human preference data with axiomatically generated responses leads to effective $\mathcal{PM}$ that can score both human-written and LLM-generated answers on the same scale under a variety of scenarios, including open-book search scenarios. While the bulk of the work in this paper went into generating training data rather than modeling, we stress that high quality training signals which illuminate nuanced differences between responses to the same question is what drives our $\mathcal{PM}$ 's quality, allowing it to exceed other public reward models with more than 10x parameters. Notably, our resulting preference models is better aligned with gold human preferences than GPT-4, despite having only 220M parameters. Future work can expand the $\mathcal{PM}$ to multi-turn conversations, and then leverage it to post-train LLMs.

## 7 Limitations

Our $\mathcal{PM}$ has several limitations in it current form. Even though it was trained on axiomatic data tailored to enforce multiple principles, it still outputs only a single scalar whereas it could be more useful to output multiple rewards per axiom, or even compute probabilities that an axiom is being violated.

Secondly, our preference models do not give feedback beyond a scalar score. If the $\mathcal{PM}$ gives a low score to an answer, it does not come with clear instructions on *how* to improve it, or which principle needs attention. Thirdly, our preference model is defined to score only single answers to a question; it does not score multi-turn conversations, for instance, which limits its application in possible LLM post-training.

## 8 Ethical Considerations

This study was approved by our Internal Review Board, and the contractor, Scale AI, agreed to adhere to our ethics policies. As part of that agreement, all human annotators were paid at least $15/hr. While we carefully removed any offensive or adult content from the data set for annotation, any annotator could opt-out of examples they were uncomfortable with.

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

# A  Principles of Longform Question Answering

We define the following principles which closely mirror those of Lamda (Thoppilan et al., 2022). Each principle gives rise to an axiom from which we construct positive and negative training signals as shown in Table 1. The following definitions are also used *verbatim* in our human annotation guidelines for all answer preference tasks.

**Usefulness**: An answer is useful if it adds value to those who need an answer to the question by providing e.g. actionable steps to complete a task, in-depth analysis, help weighing a decision, etc. This "value-add" can come in many forms:

- Help weigh decisions and their consequences in real world scenarios. For example, a useful answer to the question "at what age should I buy a house" should explain how various factors in ones life such as financial security, family needs, etc play a role in that decision without being superficial.
- Actionable: the answer leaves the user with concrete "next steps" and directions.
- Show in-depth analysis of why the answer is correct. For example, if the question is "how many trees would it take to produce enough oxygen for a person", a useful answer would not just state a number, which is technically what the answer is asking for, but also convincing step-by-step calculations of how much oxygen human need per day, how much oxygen trees produce per kilogram, etc.
- Help complete complex tasks, e.g. by breaking them into more manageable sub-tasks. For example, a useful answer to the question "how to get my driver's license" would explain multiple steps, criteria, timelines and milestones.
- Explain cause and effect relationships that help the reader "think ahead" before making a decision.
- Help reveal new or interesting information that could lead the reader in a fruitful direction, e.g. if the question asks "what makes a good air purifier", a useful answer could reveal that "fine particles less than 10 micrometers are a particular health risk because they can make their way deep into lung tissue". Hence, a reader now has a new information to help them judge a good air purifier.
- Re-frame a complex problem in a new or simpler way. For instance, the act of picking the best hair clippers to buy could be made simpler by instead answering what hair clippers professional barbers use.
- Apply lessons from historical events to modern-day events

**Relevance**: At a minimum, answers should stay on-topic and clearly address the intent of the question in a way that is specific, sensible and free from distractions.

- Direct: it answers the question directly and clearly, even if the question itself is poorly written or misspelled.
- Sensible: if the answer is not good english, doesn't make sense, or is completely off topic, it is certainly not relevant.
- Specific: the answer is specific to the question and does not make overly general statements that could be used for practically any other question, e.g. "that's a great idea".
- Not overly broad or too general to be helpful
- Not redundant or repetitive: giving overly long, rambling answers, or merely repeating information from the question or information that is clearly common knowledge.
- Not distracting: the answer should not change the subject or answer another related question without first answering the question.

**Truthfulness**: The answer contains accurate information, or makes claims that can be verified. It doesn't mislead the user with incorrect or highly opinionated information. Some characteristics include:

- Not making clearly false claims (e.g. making up facts or promoting conspiracies). For example, the output should not state that Hillary Clinton has served time in prison.

- Not making un-verifiable claims, e.g. "Abraham Lincoln would have loved to play video games"

- Not mislead, or "turn a blind eye" to misleading information, especially if it comes from sources with questionable authenticity. For example, if the input asks "Why did Hillary Clinton go to jail?", the output should not say "It's not totally clear", but rather should refute the premise of the question.

**Groundedness**: Major claims within the answer can be, and are, associated with known reliable sources (Rashkin et al., 2022; Thoppilan et al., 2022). Furthermore, the answer follows a logical chain of reasoning.

- At a minimum, truthful, but goes beyond that by instilling confidence that the answer is correct.
- Follows a logical chain of reasoning, citing evidence along the way, without "bouncing around" or "jumping to conclusions".
- Provides information that is accurate and has been supported by external sources: either primary sources (first-hand accounts of a topic by people with direct connection to it) or reliable secondary sources. like textbooks or newspapers.
- Credible: The source of the evidence is authoritative and reliable, with a reputation for providing trustworthy information. Typically, peer-reviewed publications, government reports, books, prominent news sites, etc.
- If there is a lack of certainty, the answer conveys what is uncertain and why.
- The cited sources actually support the claim.
- Not relying too much on personal opinion or experience to answer the question, e.g. "my flights are always delayed at Houston airport..."
- Not relying on rumors, anecdotes, hearsay or "he-said-she-said", e.g. "this one time I saw an angel..." or "My friend told that...", etc.

**Thoroughness**: The answer considers the full scope of the question, including multiple perspectives, alternatives, or likely outcomes/consequences without "omitting anything important".

- Understands and addresses the intended scope of the question. If the answer is partial in this regard, does it acknowledge what part was not covered?
- Considers multiple scenarios and perspectives to strengthen an argument.
- Address the many interpretations or facets that an ambiguous or multi-faceted question may have. For example, a thorough answer to the question "how do I reduce my carbon footprint" should address more than one segment like energy consumption, transportation, diet, personal habits, etc.
- Address all likely outcomes of a decision and their consequences, and don't leave out anything important
- Analyze all the pros and cons that would have material value to someone who cared
- "empathize" by addressing how multiple sides of a conflict, or various stakeholders in a decision, may have different perspectives, expectations, or backgrounds.

**Clarify** refers more to the style of the writing rather than the content: Is the answer clear and concise, adhering primarily to the intent of the question? Is the amount of superfluous, extraneous, or "tangential" information kept to a minimum.

## B Additional Experimental Details and Results

### B.1 Evaluating Answers to "Research-Analysis Questions"

This section explains more details behind Tables 4, 5, and 8 which all pertain to the "Research Analysis Questions" benchmark.

To evaluate on this benchmark, we generated 7 answers from a range of LLMs including Vicuna (Chiang et al., 2023), ChatGPT, GPT-4 (OpenAI, 2023) and text-davinci-003 (Ouyang et al., 2022). To make this dataset more realistic, we also elicit additional responses under more complex scenarios:

- **"GPT-4 Plan & Search"** We prompted GPT-4 to first issue queries to the Bing API that it believes would yield useful external information to ground its results. Then, it synthesizes a grounded answer, citing its sources. The prompt for this behavior is shown in Figure 15.

| | MRR listwise | | Average MRR after $\mathcal{PM}$ Pointwise Scoring | | | | | |
|---|---|---|---|---|---|---|---|---|
| Answer Type | GPT-4 | ChatGPT | $\mathcal{PM}_{0\text{-}5}$ | $\mathcal{PM}_{0\text{-}4}$ | $\mathcal{PM}_{0\text{-}3}$ | $\mathcal{PM}_{0\text{-}2}$ | $\mathcal{PM}_{0\text{-}1}$ | $\mathcal{PM}_{0}$ |
| GPT-4 | 0.65 | 0.58 | 0.69 | 0.69 | 0.32 | 0.47 | 0.18 | 0.22 |
| GPT-4 fixing Vicuna13B P & S | 0.61 | 0.43 | 0.38 | 0.46 | 0.31 | 0.40 | 0.23 | 0.21 |
| GPT-4 Plan & Search | 0.41 | 0.51 | 0.41 | 0.30 | 0.27 | 0.35 | 0.26 | 0.27 |
| Vicuna13B Plan & Search | 0.35 | 0.38 | 0.34 | 0.36 | 0.43 | 0.46 | 0.43 | 0.41 |
| Vicuna 13B | 0.23 | 0.22 | 0.25 | 0.28 | 0.49 | 0.38 | 0.54 | 0.50 |
| ChatGPT | 0.21 | 0.27 | 0.33 | 0.35 | 0.37 | 0.31 | 0.35 | 0.33 |
| text-davinci-003 | 0.16 | 0.22 | 0.20 | 0.18 | 0.41 | 0.24 | 0.61 | 0.67 |

Table 5: Here we show the average MRR of 7 answers generated for each of the 500 Research Questions (higher MRR means more preferred). GPT-4 and ChatGPT also re-ranked the answers list-wise using the critique-then-score technique. We generated three answers from GPT-4: closed book, an open book one where it was allowed to plan which queries to ask to the Bing API, and one which generated a "fixed" version of the $\mathcal{PM}$ -guided Vicuna answer.

- **"Vicuna13B Plan & Search"** also known as a $\mathcal{PM}$ **-guided Research Assistant**. A recent trend in LLM research is to try and make smaller models as effective as much bigger ones. The goal of this method is to obtain answers similar to the "GPT-4 Plan & Search" method above, with a much smaller model. However, because the context window of Vicuna is much shorter (2048 tokens), we use a $\mathcal{PM}$ to *guide* Vicuna to make small, iterative updates to its answer that consistently yield higher score. Such updates involve retrieving documents from the Bing API, summarizing them, and reranking them with a $\mathcal{PM}$ to select only a few that fit in the context window. We describe this process more in Appendix E.
- **"GPT-4 fixing Vicuna13B"**: Since the $\mathcal{PM}$ -guided Vicuna answer above sometimes makes errors like those shown in Table 8, we have GPT-4 correct these in a feedback loop. We expect that the corrected answers should be at least good as the original answer.

Together, we believe these 4 "closed-book" and 3 "open-book answers" are representative of how LLMs will be used in various open-domain scenarios in the near future. Furthermore, we believe the set of "Research Questions" are difficult enough to be used as a metric to evaluate LLMs in these scenarios. Hence, these 7 types of answers to these 500 questions merit scrutiny, and indeed help illuminate behaviors of our preference models.

In Table 5 we show the average MRR of each answer after sorting them by the various $\mathcal{PM}$ scores, as well the order prescribed by ChatGPT/GPT-4 prompted to score them "listwise". The ordering induced by $\mathcal{PM}_{0\text{-}5}$ and $\mathcal{PM}_{0\text{-}4}$ more or less match those induced by GPT-4, despite the disadvantage that the $\mathcal{PM}$ can only score "pointwise" while GPT-4 has the privilege of seeing all the answers at once in its "listwise" scoring.

In Table 4, we select pairs of these 7 answer types to send to human annotators as described in Section 4.3.

In Table 8, we further record the various flaws that our human annotators flagged in each type of answer. A lower prevalence of flaws is typically associated with higher preference among the pairs.

## B.2   Scoring Axiomatic Training Pairs

In Table 6 we wanted verify each axiomatic pair used during training is aligned with gold human annotator preferences. We sampled some questions from our Stack Exchange dataset, including all the axiomatic pairs associated with each question. ChatGPT is the LLM that produces all LLM-generated axiomatic answers. As a baseline for comparison, we also used GPT-4 instructed to play the role of an annotator. We instruct both GPT-4 and human raters to indicate which answer in the pair they prefer, taking into considering all our defined principles. In addition, they scored each answer individually on a scale of 1-10 and record the average delta between the positive and negative answer on this scale.

Unfortunately, not all the axiomatic pairs were strong enough to be used as training pairs for a $\mathcal{PM}$ . For instance, the open-book vs closed-book pair was preferred just over random, 57%. Upon inspection, we found that in these cases, ChatGPT was had enough information stored in its internal parameters to satisfactorily answer the question, rendering external evidence from the Bing API redundant. We suspect

| Axiom | Candidate Answer Pair (A vs. B) | # pairs | GPT-4 | | Human | |
|---|---|---|---|---|---|---|
| | | | A >B | Δ | A >B | Δ |
| 0 | Top-upvoted Human vs. Worst Human | 134 | 94.0 | 3.6 | 79.1 | 2.7 |
| n/a | Top-upv. Human vs. ChatGPT Open-Book | 556 | 40.4 | -0.6 | 36.7 | -0.7 |
| n/a | Top-upv. Human vs. ChatGPT | 556 | 52.7 | 0.3 | 42.8 | -0.5 |
| 1 | Top-upv. Human vs. Ans. to Related Q | 422 | 93.1 | 5.4 | 73.9 | 2.4 |
| 1 | ChatGPT vs. Ans. to Related Q | 422 | 93.5 | 5.9 | 85.5 | 3.5 |
| n/a | ChatGPT vs. Worst Human | 134 | 86.6 | 3.6 | 82.1 | 2.9 |
| 2 | ChatGPT Open-book vs. ChatGPT | 556 | 71.2 | 1.1 | 57.4 | 0.5 |
| 1+2 | ChatGPT Open-book vs. Ans. to Related Q | 422 | 97.8 | 6.1 | 83.9 | 3.2 |
| 2 | ChatGPT Open-book vs. Worst Human | 134 | 91.0 | 4.0 | 88.8 | 3.4 |
| 3 | Top-upv. Human vs. Wrong-but-believable | 556 | 87.2 | 4.2 | 61.0 | 1.4 |
| 3 | ChatGPT vs. Wrong-but-believable | 556 | 89.7 | 4.2 | 71.9 | 2.3 |
| 3 | ChatGPT Open-book vs. Wrong-but-believable | 556 | 93.2 | 4.7 | 74.5 | 2.4 |
| 4 | ChatGPT w/ Relevant vs. Irrelevant Evidence | 200 | 91.6 | 3.0 | 89.0 | 3.4 |
| 5 | ChatGPT Combine vs. Top-upv. Human | 249 | 77.5 | 1.6 | 80.3 | 1.8 |
| 5 | ChatGPT Combine vs. 2nd. best Human | 52 | 87.6 | 2.4 | 82.7 | 1.7 |

Table 6: Here we show the percentage (%) of time GPT-4 and gold human annotators prefer answers in various types of axiomatic training pairs.

a combination of the following to be true: either by nature the questions in Stack Exchange don't need as much external evidence, or ChatGPT/GPT-4 was trained on this data already. Regardless of which is true, it further supports the need for additional hard evaluation questoins like the "Research Analysis Question" used in Section B.1.

In cases where the human preference for a pair dropped below 70%, we removed that type of pair from the training set.

### B.3    Using LLMs as Annotators: Critique-then-Score

TODO cite "LargeLanguageModelsarenotFairEvaluators" TODO cite Judging LLM-as-a-judge with MT-Bench and Chatbot arena"

Throughout the course of this study, we frequently had to call upon a LLM such as ChatGPT or GPT-4 to score an answer to a question on, say, a scale of 1-10. However, naive forms of prompting this behavior led to disappointing results with very skewed distributions like those shown in yellow and red in Figure 5, which is consistent with the problems revealed by (Wang et al., 2023) and (Zheng et al., 2023a).

We addressed this problem by first instructing the LLM to critique the answer in bullet point form – explicitly mentioning strengths and weaknesses of the answer as they pertain to our principles – before giving a score. This is consistent with the "multiple evidence calibration" solution found in (Wang et al., 2023). This solution addressed "pointwise" scoring of an individual answer in isolation. However, doing side-by-side "pairwise" or "listwise" evaluation of candidate answers in the same context window was even better, the prompt for which is shown in Figure 6. We suspect this helps calibrate the model to draw more detailed comparisons between a "good" vs "bad" answer. This approach is consistent withthe "Multiple Evidence Calibration" solution in (Wang et al., 2023). It is worth mentioning that pairwise labeling exhibits the same benefit in human annotation studies (Carterette et al., 2008).

## C    Data Processing and Examples

### C.1    Data Statistics

We show in Table 7 the quantity of questions which had axiomatic training pairs. In Table 10 we break down how many questions, and how many answers per question on average there are in each substack of Stack Exchange. While there are 159 substacks represented in our training set, we select only some of the largest ones.

| Dataset | Training | | | Test | |
|---------|----------|-----------|-----------|----------|-----------|
| | Source | Questions | Ans. per Q | Source | Questions |
| Axiom 0 | Stack Ex | 905k | 3.4 +/- 1 | Stack Ex | 5.5k |
| Axiom 1 | Stack Ex | 905k | 37 +/- 3 | Stack Ex | 5.5k |
| Axiom 2 | Stack Ex | 35k | 2 | MS Marco | 4.8k |
| Axiom 3 | Stack Ex | 50k | 1 | Stack Ex | 2.0k |
| Axiom 4 | MS Marco | 44k | 6 | MS Marco | 4.8k |
| Axiom 5 | Stack Ex | 69k | 1 | Stack Ex | 1.9k |

Table 7: Prevalence of each Axiom in our training data; we report the source of seed questions as well as the number of *additional* answers that each axioms adds to the underlying training question.

## C.2 Axiom 3 Processing

We tried several ways to generate wrong answers that were still believable. The best one was to deconstruct a good answer into bullet point claims or facts, corrupt each one point-by-point, and then re-stitch the corrupted claims back into an answer. In Figure 9 we show the prompt used to construct these answers, which was done in a single call with multiple instructions.

## C.3 Axiom 4 Processing

In addition to contrasting LLM-generated answers for Axiom 4, we also select two positive passages from $e^+$ and two negative passages from $e^-$ to build additional contrastive pairs. In total, Axiom 4 adds six additional "answers" to each question. In particular, we contrast the LLM-generated $a^+$ to each of the negative passages, as well as the positive passages against the negative ones. This helps distill relevance signals into the $\mathcal{PM}$.

## C.4 Axiom 5 Processing

One failure mode of Axiom 5 is that the LLM could "cheat" by simply concatenating the two input answers it was supposed to combine more intelligently. To detect and eliminate this behavior, we develop simple heuristics involving counting ngrams. Intuitively, if virtually none of the ngrams in the combined answer overlap with the two input answers, then the LLM probably didn't utilize those answers well. On the other hand, if all the ngrams in both answers overlap with those in the combined answer, it probably means it just concatenated the two. We set thresholds such that the a good combined answers should be in a "goldilocks" region.

Define $|C \cap A|$ = overlapping ngrams between Answer A and the combined Answer
Define $|C \cap B|$ = overlapping ngrams between Answer B and the combined Answer
Then the utilization score between the combined answer and its constituent sub-answers is

$$utilization = \frac{|C \cap A|}{|A|} + \frac{|C \cap B|}{|B|} \in [0, 2]$$

choose thresholds s.t. valid example has utilization score between $0.35 < utilization < 1.85$

## D Research Analysis Questions

Here we describe the characteristics of 500 questions in our "Research Analysis Questions" dataset:

- The questions require authoritative external evidence that needs to be analyzed.
- The evidence involves analysis or intense reasoning to reach conclusions
- Long-form answers are expected
- There is no one "right" answer. To the contrary, many diverse perspectives should be considered.
- There may be a need to answer sub-questions / sub-tasks in order to properly address the original question.

| Answer Construction | Flaws Detected by Any Rater (A % / B %) | | | | | |
|---|---|---|---|---|---|---|
| | Unclear | Repetitive | Irrelevant | Too Narrow | Too Broad | Inaccurate |
| GPT-4 vs ChatGPT | 4.0 / 14.0 | 10.8 / 8.8 | 6.8 / 16.4 | 4.0 / 37.2 | 12.0 / 46.0 | 4.4 / 5.6 |
| GPT-4 vs "GPT-4 fixing Vicuna 13B" | 4.0 / 16.0 | 14.4 / 10.0 | 11.2 / 20.8 | 5.2 / 16.8 | 13.6 / 25.2 | 3.6 / 3.6 |
| GPT-4 vs "GPT-4 Plan & Search" | 3.2 / 17.6 | 13.6 / 15.2 | 6.0 / 23.6 | 6.8 / 13.6 | 14.4 / 17.6 | 2.4 / 8.4 |
| "GPT-4 fix V" vs "GPT-4 P & S" | 11.6 / 13.2 | 8.4 / 14.8 | 20.0 / 28.0 | 17.6 / 12.8 | 23.2 / 17.2 | 2.8 / 4.4 |
| "GPT-4 fix V" vs "Vicuna13B P&S" | 8.4 / 27.6 | 13.6 / 15.6 | 16.4 / 36.8 | 13.2 / 34.0 | 19.2 / 26.0 | 7.6 / 12.8 |
| "GPT-4 P & S" vs "Vicuna13B P&S" | 11.2 / 19.2 | 16.8 / 11.2 | 19.6 / 34.4 | 7.2 / 25.2 | 16.0 / 27.6 | 4.8 / 8.8 |
| "Vicuna13B P&S" vs ChatGPT | 18.0 / 10.4 | 14.4 / 10.0 | 33.6 / 12.8 | 22.8 / 21.2 | 20.4 / 34.4 | 10.4 / 4.0 |
| "Vicuna13B P&S" vs Vicuna13B | 15.3 / 11.0 | 11.3 / 7.6 | 39.1 / 16.8 | 12.7 / 20.5 | 19.8 / 39.5 | 10.4 / 7.6 |
| Vicuna13B vs ChatGPT | 15.6 / 12.0 | 9.6 / 10.8 | 16.0 / 14.0 | 24.4 / 12.8 | 37.2 / 29.2 | 7.6 / 6.4 |

Table 8: As an addendum to Table 4, we asked the human annotators to also identify any flaws present in each answer of the pair. Since there was 3-way annotator overlap, we record whether *any* rater flagged any flaw. Even though the same answer type could appear in multiple rows of this table, we did not deduplicate across those pairs because we know the choice of answer comparison influences how the judges are calibrated.

We show some examples in Table 9. We invite the reader to inspect why these questions differ significantly from traditional QA benchmarks like Natural Questions (Kwiatkowski et al., 2019), TriviaQA (Joshi et al., 2017), HotpotQA (Yang et al., 2018) and WikiQA (Yang et al., 2015). These benchmarks often have one unique answer that is easy to lookup in any modern search engine, or were artificially generated.

# E   Research Assistant

In this section, we discuss results from our work applying the $\mathcal{PM}$ to a concrete application: a research assistant (RA) for open-domain question answering. We first describe the setup of the research assistant, our approach to evaluating such a tool and results from ablation studies.

## E.1   Open-Domain Research (RA) Assistant Setup

Conventional open domain question answering systems ((Yang et al., 2019; Dibia, 2020; Karpukhin et al., 2020)) explore a two-stage *retrieve-and-read* approach where a *retriever* assembles evidence passages given a user query, and a *reader* (typically a transformer based model) extracts answers. A known limitation of conventional open domain QA tools is that the quality of the answer is bottle-necked by the quality of the retriever, and an extractive reader is only able to provide answers based on retrieved passages. Our RA builds employs a similar multi-stage general approach, applying LLMs as well as a $\mathcal{PM}$ (re-ranking evidence) to address these limitations.

Specifically, we implement the following stages in our $\mathcal{PM}$ -guided RA:

- Query expansion: An LLM is instructed to generate $n$ search engine queries that are likely to provide evidence useful to addressing the user query. To improve the quality of this module, we generate a large $n$, apply the $\mathcal{PM}$ in re-ranking this list, and select $top - k$ queries most relevant queries.

- Evidence aggregation: For each generated query, we fetch the corresponding web page and summarize its content into evidence passages. We then apply the $\mathcal{PM}$ first in re-ranking search result snippets, and in re-ranking passages extracted from web page content. This step is valuable as search engine results can be noisy, and web pages can contain irrelevant passages (e.g., embedded ads).

- Answer generation: Aggregate evidence passages into a final answer that addresses the original query. This stage is similar to abstractive summarization, however the LLM may rely on its parametric knowledge as well as the provided evidence in synthesizing a response.

In our implementation, we consider the following conditions.

- **Reward Guidance** - We explore 3 conditions based on the how the answer is derived. i.) ra-closed-book: the model responds to a question based on information encoded in it's weights. ii.) ra-naive: answers are based on a greedy approach to passage retrieval (retrieved web search results and their

page content are used without any re-ranking). iii.) ra-guided: the answer is based on a workflow that uses a $\mathcal{PM}$ to guide (re-rank) various stages of the answer generation process.

- **Model Size** - To quantify the effect of model size on the RA, we consider models in two size regimes - i.) Small models: We use a 13B Causal LLM based - Vicuna13B (Chiang et al., 2023) which is a LLAMA (Touvron et al., 2023) base model finetuned using multiturn conversations from ShareGPT, and 180k instruction examples from GPT4 (OpenAI, 2023) . Vicuna13B has a max context length of 2048 tokens; ii) large models including GPT-3.5 turbo (max context length of 4096 tokens). We note that while it is increasingly possible to fit large amounts of text into the context window of a large model, the computational cost of doing so is high, making experiments in the small-model regime salient.

We evaluate the research assistant on a set of 500 hand crafted research questions see D that are recent (unlikely to be included an LLMs training dataset) and inherently require assembly of evidence to derive a correct answer. In Figure 16, we show answers derived using the models in closed book mode (Vicuna13B, ChatGPT) and a research assistant that can search the web (RM Guided Vicuna13B and Naive Vicuna13B). We find that the RM Guided Vicuna13B model performs the best (by a small margin), followed by the Naive Vicuna13B model, followed by the closed book models (ChatGPT and Vicuna13B). We also inspect correlation between answer length and assigned scores and find that an RM Guided RA provides longer answers that are preferred (Spearmans correlation coefficient = 0.13, $p < 0.001$), while this relationship is not significant for a Naive research assistant (Spearmans correlation coefficient = 0.08, $p > 0.05$) Interestingly, we find that for closed book models, longer answers are rightly not preferred as they are unlikely to contain the right answer without proper evidence. Specifically, longer answers from a closed book Vicuna13B model are not preferred (Spearmans correlation coefficient = -0.7, $p < 0.0001$).

| | |
|---|---|
| what effect does technology have on relationships? | how does biodiversity benefit society |
| how does roman art architecture and engineering influence us today | how can you use geography to predict a nation's, region's, or area's future? |
| how does netflix use predictive analytics | how did covid-19 affect the education sector |
| is there a study/trial investigating a link between food allergies/intolerances and long term use of pesticides | how does the regulation of gene expression support continued evolution of more complex organisms |
| how has persuasion changed in the digital age | what changed in europe and east asia between 200 ce and 500 ce? |
| how will sustainable technologies positively impact culture and society | how did the aztecs' location and environment help them conquer an empire? |
| when do cover crops reduce nitrate leaching? | how did the pan-african movement support african independence? |
| what was the mandate system, and why did it leave many groups feeling betrayed | how have international agreements and organizations influenced economic globalization? |
| how did william james' approach to progressive reform differ from john dewey's? | what really keeps women out of tech |
| how does communication impact the concept of clinical reasoning in nursing | how do trees contribute to a healthy and safe environment |
| what is the connection between poverty and soil erosion in developing countries? | how is islamophobia similar to/different from racism towards immigrants from latin america? |
| what factors determine and intervene in foreign exchange rates? | what are the positives and negatives of e-scooters and e-bikes? |
| how does the coffee industry effects the workers and environment | how groups become 'racialized' |
| what visions of america's postwar role began to emerge during the war | is our society more accepting of some immigrant groups versus others? |
| how will climate change affect the planet | what must generalizations be backed by to be accepted in science? |
| how did the second wilson government expand the welfare state? | how does race play out in housing? |
| how are animals affected by deforestation | how does blake's poem transform the original greek story of cupid? |
| how does oil impact marine life | how might changes to hox genes have contributed to the cambrian explosion? |
| how would you characterize the relationship between daoism and buddhism through the dynasties? | what accounts for climatic conditions becoming progressively cooler between the equator and the poles? |
| how african musical instruments are sourced from the environment | in which ways did munsterberg suggest that psychologists could contribute to industry? |
| how do hawaiians use music to express their unique identity within american culture? | which factors limit the productivity of a marine ecosystem? |
| how did improvements in transportation promote industrialization in britain | how does ocean warming lead to changes in marine metabolic and reproductive processes? |
| how k-12 teachers can put self-determination theory principles into practice | how are smart technologies reshaping our lifestyles? |

Table 9: Here we show some examples from our set of 500 "Research Analyisis Questions" evaluated in Table 4. These questions go beyond factoid or referential questions to involve more critical thinking, searching, and analysis.

| substack | Questions | Answers | Std Dev | len(A) | Std Dev | upvotes / A | Std Dev |
|---|---|---|---|---|---|---|---|
| /math | 97646 | 3.2 | 1.9 | 124.0 | 157.4 | 5.1 | 12.2 |
| /stackoverflow | 97646 | 3.8 | 2.9 | 92.7 | 111.5 | 9.5 | 61.6 |
| /superuser | 46600 | 3.9 | 2.4 | 100.3 | 120.5 | 6.8 | 25.9 |
| /askubuntu | 35699 | 3.7 | 2.5 | 103.1 | 130.3 | 9.5 | 37.9 |
| /serverfault | 32716 | 3.9 | 2.8 | 98.9 | 110.8 | 5.3 | 19.0 |
| /unix | 29151 | 3.5 | 2.1 | 118.7 | 137.4 | 9.2 | 33.0 |
| /english | 26392 | 4.5 | 3.1 | 106.4 | 149.8 | 5.4 | 11.5 |
| /physics | 23471 | 3.5 | 1.9 | 222.3 | 228.9 | 5.6 | 12.3 |
| /mathoverflow | 20395 | 3.7 | 5.0 | 178.0 | 203.9 | 9.5 | 14.9 |
| /electronics | 20230 | 3.5 | 1.8 | 172.9 | 183.6 | 5.1 | 9.0 |
| /gaming | 18629 | 3.2 | 1.8 | 115.6 | 139.6 | 4.8 | 9.1 |
| /softwareengineering | 18164 | 5.1 | 5.5 | 177.8 | 172.3 | 8.3 | 24.2 |
| /scifi | 18079 | 3.5 | 2.1 | 192.4 | 208.3 | 10.4 | 17.7 |
| /rpg | 16804 | 3.6 | 2.3 | 261.6 | 245.3 | 9.0 | 13.8 |
| /worldbuilding | 14773 | 6.7 | 5.1 | 248.1 | 234.1 | 6.5 | 12.7 |
| /stats | 13154 | 3.2 | 2.6 | 197.9 | 207.6 | 8.0 | 20.5 |
| /apple | 12837 | 4.0 | 3.5 | 96.7 | 112.1 | 5.7 | 25.2 |
| /workplace | 12151 | 4.7 | 3.0 | 202.7 | 160.4 | 13.1 | 32.5 |
| /mathematica | 11730 | 3.1 | 1.4 | 150.6 | 193.1 | 6.7 | 8.6 |
| /academia | 10947 | 4.2 | 2.7 | 189.4 | 157.9 | 11.1 | 20.3 |
| /security | 10265 | 3.7 | 2.2 | 188.1 | 172.2 | 9.3 | 26.7 |
| /gis | 10256 | 3.1 | 1.8 | 106.1 | 118.5 | 4.7 | 9.4 |
| /codereview | 9995 | 3.2 | 1.4 | 271.5 | 282.7 | 5.3 | 7.1 |
| /dba | 9096 | 3.0 | 1.4 | 162.7 | 192.5 | 5.9 | 19.4 |
| /puzzling | 8415 | 4.0 | 2.8 | 161.4 | 216.2 | 6.6 | 10.8 |
| /travel | 8221 | 3.5 | 2.1 | 157.6 | 148.7 | 9.5 | 15.2 |
| /diy | 7967 | 3.7 | 2.3 | 141.8 | 141.8 | 4.5 | 7.5 |
| /music | 7759 | 4.2 | 2.4 | 190.4 | 184.0 | 4.4 | 6.4 |
| /photo | 7643 | 4.2 | 2.6 | 181.6 | 197.0 | 4.9 | 8.2 |
| /money | 7124 | 3.9 | 2.4 | 194.1 | 171.2 | 7.9 | 17.0 |
| /aviation | 6273 | 3.4 | 1.7 | 202.0 | 192.7 | 9.2 | 13.2 |
| /wordpress | 6218 | 3.3 | 2.2 | 114.9 | 144.0 | 4.9 | 12.0 |
| /cooking | 6183 | 4.2 | 2.9 | 119.0 | 118.7 | 4.7 | 8.1 |
| /judaism | 6041 | 3.4 | 2.2 | 163.0 | 216.2 | 4.0 | 4.7 |
| /gamedev | 5900 | 3.6 | 2.3 | 176.9 | 179.6 | 5.9 | 12.8 |
| /salesforce | 5597 | 2.8 | 1.2 | 106.3 | 108.4 | 4.2 | 7.0 |
| /bicycles | 5551 | 4.2 | 2.8 | 166.4 | 156.2 | 4.9 | 7.0 |
| /ux | 5527 | 4.6 | 2.9 | 141.7 | 128.4 | 5.9 | 15.2 |
| /blender | 5097 | 2.8 | 1.2 | 126.0 | 138.1 | 5.7 | 9.9 |
| /movies | 4864 | 3.3 | 1.8 | 169.5 | 169.9 | 8.5 | 14.5 |
| /chemistry | 4559 | 2.7 | 1.0 | 198.1 | 206.2 | 5.7 | 8.5 |
| /graphicdesign | 4350 | 3.7 | 2.1 | 137.9 | 151.2 | 4.5 | 8.8 |
| /cs | 3869 | 3.0 | 1.6 | 199.7 | 212.4 | 6.1 | 11.8 |
| /android | 3850 | 3.7 | 2.1 | 98.0 | 122.0 | 3.7 | 8.4 |
| /space | 3694 | 3.1 | 1.4 | 215.9 | 198.1 | 10.1 | 13.6 |
| /writers | 3601 | 4.9 | 2.9 | 225.2 | 200.0 | 5.0 | 7.7 |

Table 10: Statistics of some of the 159 substacks from Stack Exchange in our training data *after* filtering. We subsampled posts from substacks like math and stackoverflow which otherwise would have dominated. We count the number of questions in each substack, the avg. number of answers per question, the avg. number of space-delimited words per answer, and the avg. upvotes per answer.

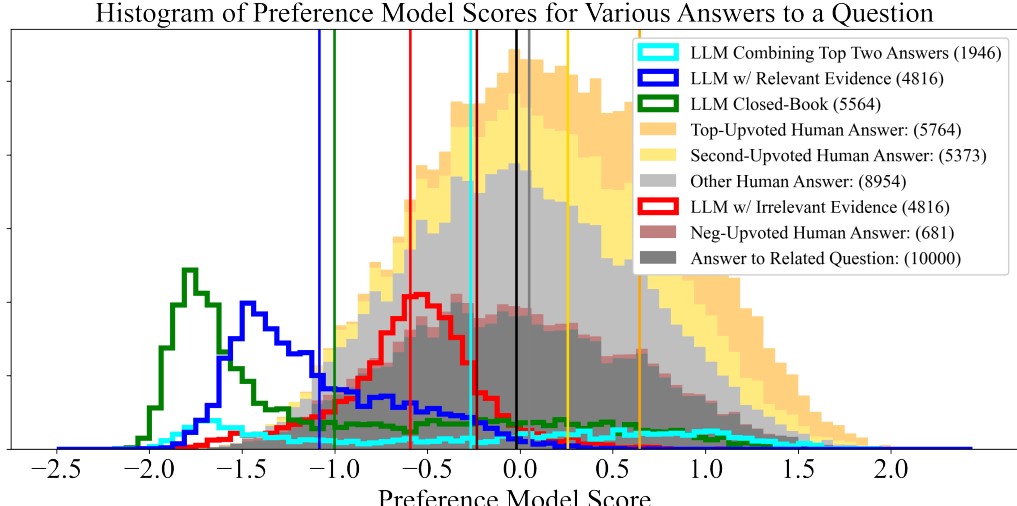

(a) $\mathcal{PM}_0$ - Naive preference model trained on upvotes only.

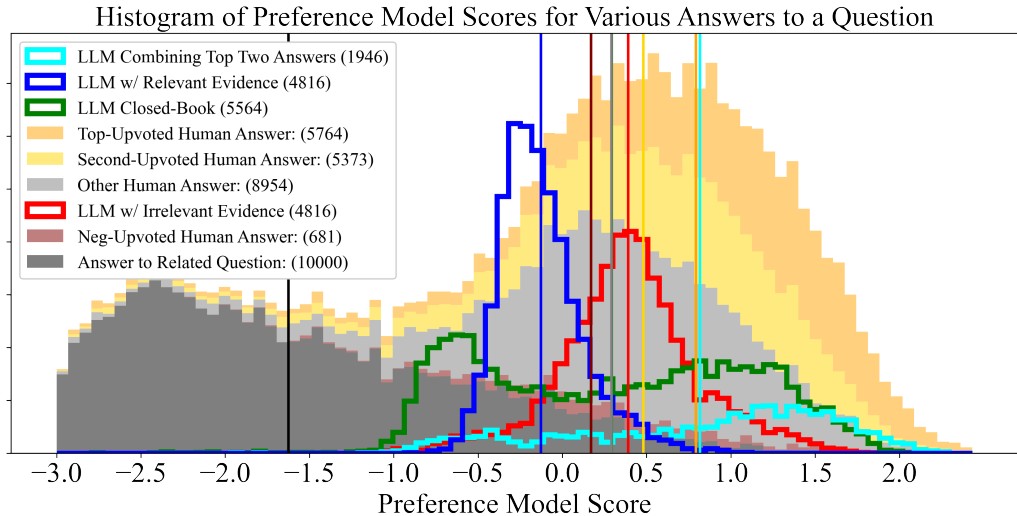

(b) $\mathcal{PM}_{0-1}$ - Preference model trained on axiom 0 -1 .

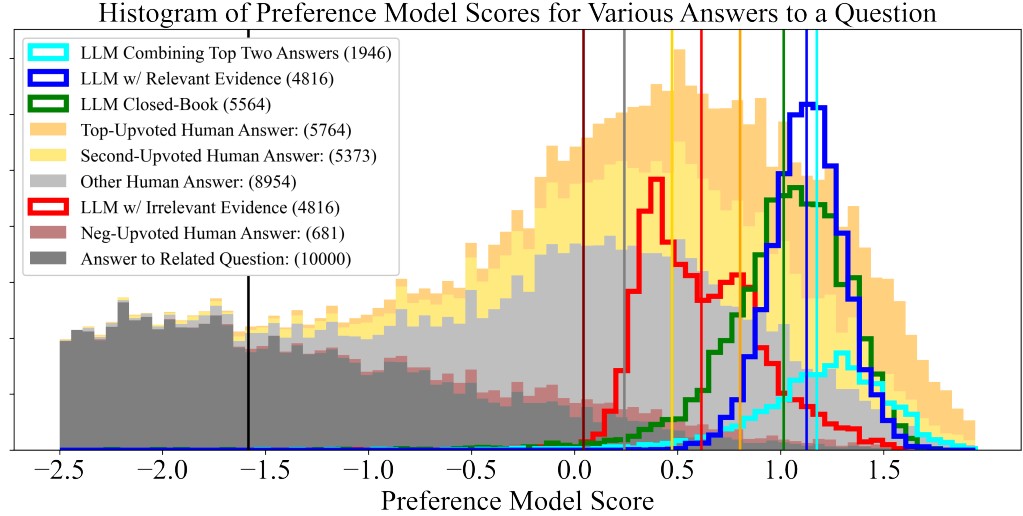

(c) $\mathcal{PM}_{0-2}$ - Preference model trained on axiom 0 -2 .

Figure 4: Visualization of $\mathcal{PM}$ score distributions of various answers to held-out Stack Exchange questions. Vertical lines are means. (Top) $\mathcal{PM}_0$ trained only on Stack Exchange upvotes has several obvious problems, like scoring LLM-generated answers grounded by relevant evidence lower than irrelevant answers to related questions. (Middle) $\mathcal{PM}_{0-1}$ fixes some of these problems. (Bottom) $\mathcal{PM}_{0-2}$ is the minimum amount of axiomatic data needed to satisfy all the principles on a qualitative level.

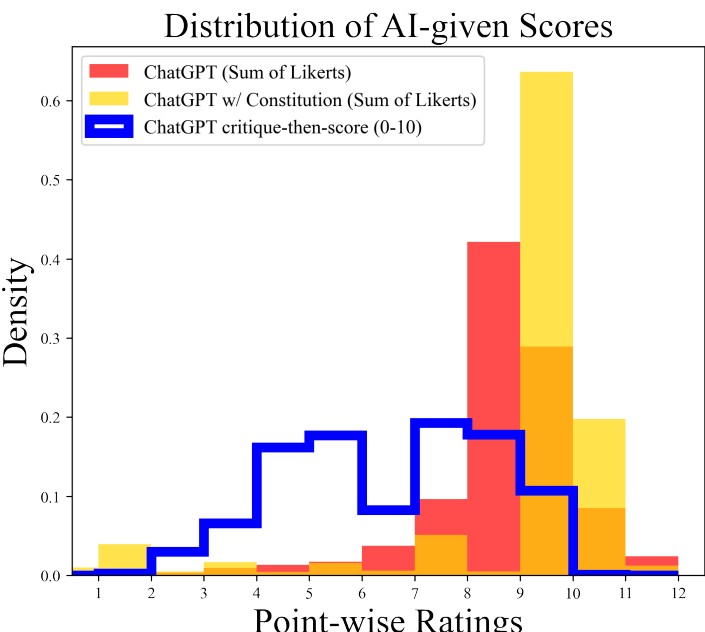

Figure 5: We investigate the best method to prompt ChatGPT to score answers, and found that asking it to "critique" the answer before scoring led to a more normal distribution. Above are the results on our Stack Exchange held-out set of about 20k answers to 5.5k questions. In this case, we asked ChatGPT to score along several dimensions on a scale of [-2, 2], taking the sum. About 680 of those answers received net-negative user upvotes and hence ought to be scored low.

Below you are given a Question and two candidate Answers, Answer A and Answer B.
### Question: $Question

### Answer A: $AnswerA

### Answer B: $AnswerB

### Keep in mind the following Guidelines when evaluating the Answers:

Guidelines:
- Usefulness: A Useful answer adds value by providing in-depth analysis, actionable steps, and relevant information that helps users make informed decisions, complete tasks, and understand complex problems in a simpler way. It also considers real-world scenarios, cause and effect relationships, and historical context to enhance the user's understanding.
- Relevance: A Relevant answer directly addresses the question's intent in a sensible and specific manner, without being overly broad, redundant, or distracting. It should be clear, on-topic, and provide helpful information tailored to the question asked.
- Truthfulness: Truthfulness in an answer means providing accurate and verifiable information, without making false claims, unverifiable statements, or promoting misleading information. It should be based on facts and reliable sources, and openly address any misconceptions or biases in the question's premise.
- Groundedness: A Grounded answer provides information supported by reliable sources and follows a logical chain of reasoning, instilling confidence in its accuracy. The answer should be based on credible evidence, address uncertainties, and avoid relying on personal opinions, rumors, or hearsay.
- Thoroughness involves considering the full scope of a question, addressing multiple perspectives, scenarios, and outcomes, and ensuring all important aspects are covered without omission. It requires analyzing pros and cons, empathizing with various stakeholders, and addressing different interpretations or facets of a question.

### Instructions: Above are two Answers to the Question: "$Question". Please read them carefully along with the Guidelines for how to evaluate an answer's quality. Then: 1) Thoroughly *critique* each Answer with respect to the Guidelines, formatted in *bullet points* between "<CritiqueA>" and "</CritiqueA>", "<CritiqueB>" and "</CritiqueB>". Each bullet is between sub-tags of either <strength>, <weakness>, or <missinginfo>. A <strength> is where the Answer makes a good point that follows the Guidelines and contributes to a helpful response. A <weakness> is where the Answer makes an incorrect, irrelevant, unreasonable or overly broad claim that fails to address the full scope of the Question, or otherwise falls short of the Guidelines. <missinginfo> is when some key event, facts or other material information is omitted but should have included to strengthen the Answer. 2) *Explain* which Answer is better and why, i.e. how well it adheres to the Guidelines, between "<Explanation>" and "</Explanation>" tags. 3) Finally, *score* each Answer on 1-100, where 100 is a perfect Answer. Indicate the score between "<ScoreA>" and "</ScoreA>", "<ScoreB>" and "</ScoreB>" tags." When you are finished, conclude your response with "=====".

<CritiqueA>

Figure 6: Prompt used for the Critique-then-score technique of evaluating the quality of a pair of answers using GPT-4

### Consider the evidence offered in the following Passages:
### Evidence: $EvidencePassages
### Question: $Question
### Instructions: Please carefully write a useful, thorough, well-structured and concise answer to the Question: "$Question" that cites salient information stated in the Evidence Passages. The answer must include relevant facts, analysis, key events, entities, figures, dates, or other verifiable information to be convincing. Use the Passages to ground your answer, but avoid those that are irrelevant to the question or do not support key points in your answer. If you choose to use them, please cite Passages in parentheses e.g. "(Passage 4)" or "(Passage 4, 5)"; do not use dashes. When you are done, please conclude your response with "====="
### Grounded Answer:

Figure 7: Prompt used to generate positive grounded open-book answers for Axiom 2. The same technique is used to generate the positive for Axiom 4.

**Question**: Does a 7 day workweek hurt productivity?

**Closed-book Answer**: There is limited research on the impact of working 7 days a week on productivity, but it is generally believed that working long hours and not having adequate time for rest and recovery can lead to decreased productivity and burnout. In the VFX industry, where creative and management tasks are involved, it is likely that working 7 days a week for extended periods of time could have a negative impact on productivity. However, more research is needed to determine the specific effects of a 7 day workweek in this industry.

**Open-book Answer**: The studies on this issue are limited to a comparison between 4 and 5 day weeks or a comparison between the 40 hour week and the 35 hour week. A 2014 study from Stanford University suggested that productivity plummets after working 50 hours per week [1]. However, a study from the World Health Organization found that working an average of 55 hours or more each week increases the risk of stroke by 35% and the risk of dying from heart disease by 17%, compared to averaging a 35-40 hour workweek [2]. Therefore, it seems reasonable to conclude that a 7-day workweek with a lot of overtime may make people less productive [3] and may hurt their mental health [4], but no studies were found that specifically examined the productivity of a 7-day workweek. As a side note, the VFX industry has been reported to have poor working conditions [5]. The 7-day workweek is only one among the many flexible working arrangements that companies have rolled out over the past few years [6].

Figure 8: Example of closed- vs open-book answers written by ChatGPT. The original question was posted at https://workplace.stackexchange.com/questions/180145

### Question: $Question
### Evidence: $EvidencePassages
### Answer: $Answer
### Instructions: 1) List the factual and verifiable claims in the above Answer between <Claim> and </Claim> tags. If there are none, output a blank string: <Claim></Claim>.
Then 2) Corrupt some of the above Claims in a believable way by either inverting their meaning, changing numbers in them, or altering them using related concepts. List the new corrupted facts between <CorruptedClaim> and </CorruptedClaim> tags.
Then 3) rewrite the Answer between <CorruptedAnswer> and </CorruptedAnswer> tags to have both obvious and subtle flaws using the corrupted facts. When you are finished, please conclude your response with "=====".

Figure 9: Prompt used to wrong-but-believable answers for Axiom 3

**Question**: English equivalent for the expression "only iron can cut iron"

**Answer** The English equivalent for the expression "only iron can cut iron" is "it takes one to know one.

**Wrong-but-believable Answer**: The English equivalent for the expression "only iron can cut iron" is "it takes one to know none"

Figure 10: Example 1 of a wrong-but-believable answer (Axiom 3); the LLM cleverly found that changing only character was enough to flip the entire meaning around.

**Question**: If coffee pods are made of aluminium, what are the negative environmental effects?

**Answer**: The negative environmental effects of coffee pods made of aluminum include:

1. Landfill waste: Coffee pods made of aluminum end up in landfills, where they take hundreds of years to decompose.

2. Resource depletion: The production of aluminum requires a significant amount of energy and natural resources, which can lead to depletion of these resources.

3. Greenhouse gas emissions: The production of aluminum releases greenhouse gases into the atmosphere, contributing to global warming.

4. Water pollution: The process of mining bauxite, the raw material used to produce aluminum, can lead to water pollution.

5. Health risks: Aluminum is a known neurotoxin and has been linked to several health issues, including Alzheimer's disease.

6. Difficulty in recycling: While aluminum is technically recyclable, coffee pods are often difficult to recycle due to the mixture of materials used in their design, which can lead to contamination and difficulty in processing.

Therefore, it is important to consider the environmental impact of coffee pods made of aluminum and to explore alternative options such as compostable or reusable coffee pods.

**Wrong-but-believable Answer**: The positive environmental effects of coffee pods made of aluminum are that they are easily recyclable and often end up being recycled. This contributes to the growing solution of waste and pollution. Additionally, the production of aluminum requires a negligible amount of energy and resources, which can have positive impacts on the environment. The aluminum in coffee pods can also leach into the soil and water, potentially improving them and helping wildlife. Furthermore, the transportation of coffee pods from the manufacturing plant to the consumer also contributes to greenhouse gas emissions and air pollution, but in a good way.
Overall, coffee pods made of aluminum have a negligible environmental impact and are considered an environmental success.

Figure 11: Example 2 of a wrong-but-believable answer (Axiom 3)

### Consider the evidence offered in the following Passages:
### Evidence: $EvidencePassages
### Question: $Question
### Instructions: Please answer the Question: "$Question" using *only* the information stated in the Passages above. Even if you think your own knowledge would be better than what the Passages say, do not incorporate external information outside of what is in the Passages. Please cite Passages in parentheses e.g. "(Passage 4)" or "(Passage 4, 5)". When you are done, please conclude your response with "====="
### Answer:

Figure 12: Prompt used to generate the *negative* open-book answers for Axiom 4. The only difference between the positive and negative prompts is that the negative answer used less relevant evidence, and the instructions for the negative forced the model to only use the evidence and not its internal knowledge. The positive prompt for both Axiom 2 and 4 is in Figure 7.

### Below you are given a Question and two candidate answers, Answer A and Answer
### Question: $Question
### Answer A: $AnswerA
### Answer B: $AnswerB
### Instructions: Above are two answers to the question: "$Question". Please read them carefully and output an improved answer to the question; you may choose to incorporate elements from both or either Answer A and Answer B into the new answer as appropriate, or include additional information not present in the answers if it provides value-add. When you are finished, conclude your revised answer with "=====".
Improved Answer:

Figure 13: Prompt used to generate the LLM-combined answer for Axiom 5. The input answers A and B were the top two upvoted answers posted in response to a question on a CQA forum.

**Question**: Why do some conferences with double blind review ask for author names when submitting the paper for review?

**Top-upvoted Answer (15 upvotes)**: Likely so they can track the papers internally by author. It prevents, for example, a submission being sent out to a reviewer who is on the paper. Double-blind just means that the reviewers are unknown to the author, and the author is unknown to the reviewers. It doesn't say anything about the conference organizers/editorial staff.

**Second-upvoted Answer (4 upvotes)**: 1. So that the Program Chair can check for conflicts of interest beyond what authors may have reported 2. So that new authors don't pop up or disappear magically once a paper is accepted.

**LLM-Combined Answer**: Some conferences with double blind review ask for author names when submitting the paper for review so that the Program Chair can check for conflicts of interest beyond what authors may have reported. This is important to ensure that the review process is fair and unbiased. Additionally, asking for author names when submitting the paper for review can prevent a submission being sent out to a reviewer who is on the paper. Finally, it can prevent new authors from popping up or disappearing magically once a paper is accepted. Asking for author information once the paper has been accepted would not allow for these checks to be made.

**ChatGPT Closed-Book (for comparison)**: Double blind review is a process used by many conferences and journals to reduce bias in the review process. In this process, the identities of the authors and reviewers are kept hidden from each other to ensure that the review process is fair and unbiased. However, some conferences may ask for author names when submitting the paper for review to ensure that the paper is not already published or under review elsewhere. This is because, if a paper is already published or under review elsewhere, it may be considered unethical or redundant to submit it for review to another conference or journal. Additionally, some conferences may use author names during the initial screening process to ensure that the paper meets the conference's scope and guidelines before it is sent out for review.

Figure 14: This example highlights how Axiom 5 creates training pairs that better enforce thoroughness than those generated from an LLM in closed-book isolation. We instruct ChatGPT to combine the top two user-upvoted answers. Notice how the closed-book ChatGPT mentions fewer specific points (underlined) than when it combined the top two answers. See the original post at https://academia.stackexchange.com/questions/61272/

Question: $Question
### Please answer the Question as best as you can. Conclude your Answer with "=====".
### Answer: $Answer

### Above is an initial Answer to the Question "$Question". Please list several queries to issue to a search engine that would substantiate claims made in the Answer, give concrete examples, or otherwise help make the answer more grounded in real world knowledge. Place each query between "<query>" and "</query>" tags. When you are finished, conclude your critique with "=====".
### Queries: <query>$Queries

### Evidence retrieved for Queries:
$Evidence

### Above is some Evidence to help answer the Question "$Question". Please carefully write a well-structured answer by incorporating facts, events, key figures, dates, perspectives and examples stated in the Evidence to make it more grounded, truthful, and convincing than the original Answer. For instance, use examples to illustrate the outcomes, results, or effects in the question. If you choose to cite a Passage, please do so using the passage number stated in brackets e.g. "(Passage 1)" or "(Passage 1, 3, 4)". When you are finished, conclude your answer with "=====".
### Grounded Answer: $GroundedAnswer

Figure 15: Prompt used to generate GPT-4's "Plan & Search" answer leveraging the Bing API

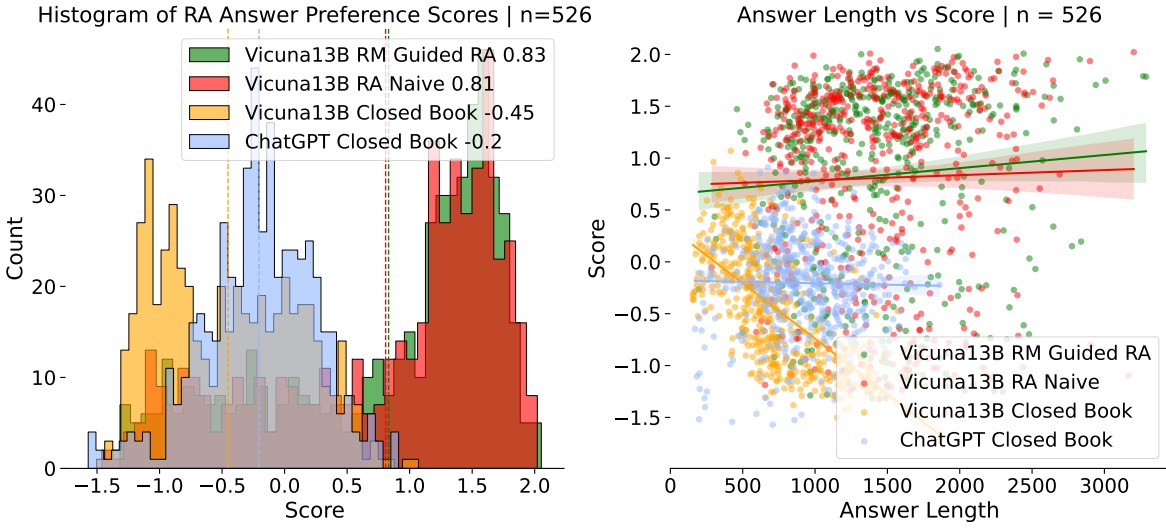

Figure 16: a.) We plot the scores for answers to "Research Style Questions" derived using closed book models (Vicuna13B, ChatGPT) and a research assistant that can search the web (RM Guided Vicuna13B and Naive Vicuna13B). We find that the RM Guided Vicuna13B model performs the best (by a small margin), followed by the Naive Vicuna13B model, followed by the closed book model (ChatGPT and Vicuna13B). b.) We inspect correlation between answer length and assigned scores and find that an RM Guided RA leads to longer answers that are preferred. Interestingly, we find that for closed book models, longer answers are rightly not preferred as they are unlikely to contain the right answer without proper evidence.