# OpenReview forum: "Axiomatic Preference Modeling for Longform Question Answering"
_EMNLP/2023/Conference — EMNLP 2023 Main_

### Official Review · Reviewer_k1y7 · 2023-07-28

**Soundness:** 3

**Excitement:**

3: Ambivalent: It has merits (e.g., it reports state-of-the-art results, the idea is nice), but there are key weaknesses (e.g., it describes incremental work), and it can significantly benefit from another round of revision. However, I won't object to accepting it if my co-reviewers champion it.

**Paper Topic And Main Contributions:**

This paper argues that reward models (RMs) in RLHF often lack direct knowledge of the reasons or principles of preference annotations. Therefore, the authors design five principles, both human-based and LLM-based, to guide RMs to better align with human preferences in the field of longform question answering. Overall, this paper might contribute to the "NLP engineering experiment".

**Questions For The Authors:**

- Question A: Since some preference signals are created using ChatGPT, what if ChatGPT is used as a baseline and compared to your model, especially without using human reference data?
- Question B: How did you choose the threshold tq in line 243, could you provide some details?

**Reasons To Accept:**

- Based on well-defined principles, this paper shows the effectiveness of the preference model in extensive experiments. The model with only 220M parameters even exceeds the capabilities of GPT-4.
- The proposed framework can be extended to other related fields, and can potentially contribute to RLHF.

**Reasons To Reject:**

- The authors propose that existing RM-based LLMs are trained to regress human-preference scores without clear knowledge of why they made that decision. However, the main idea of this paper is to augment human preference data with LLM-generated pairs. The reliability of this approach is somewhat unconvincing.
- In Table 2, the results show that some signals are not necessary or even hurt the model in some datasets, especially the results of PM (0-5). This requires more analysis of the impact of different signals on the model, although the evaluation for some axioms are provided in Table 3.

**Reproducibility:**

3: Could reproduce the results with some difficulty. The settings of parameters are underspecified or subjectively determined; the training/evaluation data are not widely available.

**Reviewer Confidence:**

3: Pretty sure, but there's a chance I missed something. Although I have a good feel for this area in general, I did not carefully check the paper's details, e.g., the math, experimental design, or novelty.

---

> ### Author Rebuttal · Authors · 2023-08-29
>
> We would like to thank Reviewer k1y7 for their detailed comments and feedback.
>
> We restate the key findings of our work:
> 1. that the composition of training pairs for preference models is of paramount importance (training data mix) and
> 2. In our new experiments, even open source preference models with > 10x parameters have structural flaws that limit their performance (shown below) – flaws that can be fixed by training much smaller models on our axiomatic pairs.
>
> We are able to make these claims because our study is the first to earnestly measure preference models across a comprehensive suite of datasets with a variety of labels (user upvotes, gold human annotations, and GPT-4 scores). We refer to the following **updated version of Table 2** in our rebuttal below.
>
>
> |                  |      StackX     |          |      r/ELI5     |          |    r/Science    |          |    r/History    |      |     MS Marco    |          |    WebGPT    |
> |------------------|:---------------:|:--------:|:---------------:|:--------:|:---------------:|:--------:|:---------------:|:----:|:---------------:|:--------:|:------------:|
> |   Avg. Ans per Q | 3.6 pos, 40 neg |          | 4.6 pos, 43 neg |          | 6.5 pos, 42 neg |          | 5.3 pos, 47 neg |      | 1.1 pos, 1k neg |          | 1 pos, 1 neg |
> |           Metric |       MRR       |   NDCG   |       MRR       |   NDCG   |       MRR       |   NDCG   |       MRR       | NDCG |       MRR       |   NDCG   |   Accuracy   |
> |      length(Ans) |       15.0      |   35.4   |       6.2       |   27.6   |       7.7       |   30.1   |       15.0      | 37.1 |       n/a       |    n/a   |     56.7     |
> | **OpenAsst-RM 6.7B** |       25.0      |   44.6   |       12.7      |   34.7   |       15.4      |   38.1   |       24.4      | 46.1 |       4.0       |   17.3   |   **76.5**   |
> | **StackLlama RM 7B** |       26.8      |   45.1   |       8.3       |   30.6   |       10.3      |   33.3   |       9.8       | 33.1 |       3.4       |   16.1   |     56.1     |
> | GPT-4 (listwise) |       45.5      |   62.1   |       39.6      |   59.9   |       35.1      |   56.4   |       37.8      | 60.4 |       n/a       |    n/a   |      n/a     |
> |     PM_0 T5-base |       31.2      |   48.6   |       11.1      |   32.6   |       14.8      |   37.0   |       24.0      | 44.5 |       3.9       |   16.9   |     51.1     |
> |   PM_0-1 T5-base |       64.3      |   78.8   |       54.5      |   75.2   |       53.2      |   75.4   |       63.1      | 84.3 |       16.1      |   30.6   |     55.7     |
> |   PM_0-2 T5-base |       65.5      |   79.8   |       55.1      |   76.3   |       51.9      |   74.6   |       61.4      | 83.1 |       9.7       |   25.6   |     57.6     |
> |   PM_0-3 T5-base |       65.3      |   79.5   |       55.0      |   76.0   |       51.4      |   73.9   |       61.1      | 82.8 |       9.4       |   23.7   |     55.4     |
> |   PM_0-4 T5-base |       65.8      |   80.0   |       54.0      |   75.2   |       51.1      |   74.0   |       61.2      | 83.0 |     **25.0**    | **39.3** |     58.6     |
> |   PM_0-5 T5-base |       64.6      |   79.2   |       53.6      |   75.0   |       51.6      |   74.3   |       61.7      | 83.3 |       23.1      |   37.4   |     58.1     |
> |  PM_0-5 **T5-large** |     **66.4**    | **80.8** |     **55.9**    | **77.0** |     **55.4**    | **77.2** |     **64.0**    | **85.4** |       24.3      |   38.9   |     59.1     |
>
> # Reason To Reject Point 1
> > "the main idea of this paper is to augment human preference data with LLM-generated pairs.”
>
> We would like to clarify that the central focus of our paper is not merely on augmenting data with additional generated pairs. Instead, our primary objective is to employ an 'axiomatic' framework to teach the preference Model (PM) about various dimensions of quality. The data generation process serves as a means to operationalize this concept, facilitating the efficient creation of differentiated pairs that vary along specific attributes such as groundedness and truthfulness.
>
> To strengthen the credibility of our results, we conducted an additional experiment in which we compared our PM_0-5 model to two strong baselines. The first is **LAION Open-Assistant 6.7B-parameter** preference model, and the second is **HuggingFace's Stack-Llama 7B-parameter** reward model. **Remarkably, our PM_0-5 model, which has only 220M parameters, outperformed both of these significantly larger models**. This is especially notable given that the Stack-Llama model was also trained on Stack Exchange data.
>
> # Reason To Reject Point 2
>
> > ”the results show that some signals are not necessary”
>
> The focus of the ablations in Table 2 is to track the composition of axiomatic datasets to support the conclusion that more axiomatic signals generally lead to better PMs.
>
> However, as Reviewer 3 points out, this ablation does reveal a few discrepancies, such as PM_0-5 being slightly lower than PM_0-4 (64.6 vs 65.8 resp. on StackX MRR) in Table 2 when using T5-base as the underlying PM. These discrepancies are relatively small compared to the huge jump we see over the baselines like PM_0 (31.2 StackX MRR). We have reason to believe that **model capacity can explain these variations**: our additional experiment in the table above replacing T5-base with **T5-large** in PM_0-5 to show that **larger capacity preference models can better absorb the different data distributions**.
>
> We want to emphasize that the discrepancies we reveal are not a bad thing (nor a reason to reject the paper), since raising awareness of them will benefit others in the community as they train their preference models. Besides, we overcome the discrepancies with more model capacity.
>
> # Question A
> > ”what if ChatGPT is used as a baseline and compared to your model”
>
> Instead of ChatGPT, we used the more powerful GPT-4  as a baseline to score and re-rank answers in Table 2 to compare against our PMs. It is a good point that we should run an experiment “without using human reference data” which we interpret as a PM_1-5 (no user upvotes from StackExchange in Axiom 0).  We will include this in the camera ready.
>
> # Question B
> > ”How did you choose the threshold tq in line 243”
>
> Excellent question: we choose the t_q threshold to indicate that any two questions whose dot product is > t_q have the same intent. We selected 50 questions {q} at random and retrieved the 100 nearest neighbors for them q’ \in knn(q), and manually annotated which of the top 100 are “co-intent”, recording their dot product scores in a histogram. In this histogram, visually, t_q = 20 was the vertical line which had the best separation (co-intent were to the right, and non-co-intent were left). We will include a description of this in the camera ready.
>
> # Errata
> Note, in the submitted paper, we accidentally entered the incorrect scores for PM_0-5 T5-base for StackX and Reddit numbers in Table 2. The corrected numbers are presented in the table above. The changes were small (e.g. Stack X MRR was incorrectly entered as 63.0 and the corrected number is 64.6, r/ELI5 MRR went from 53.4 to 53.6). The model itself did not change, we just misread a row in our spreadsheet.

---

### Official Review · Reviewer_83C4 · 2023-07-29

**Soundness:** 4

**Excitement:**

5: Transformative: This paper is likely to change its subfield or computational linguistics broadly. It should be considered for a best paper award. This paper changes the current understanding of some phenomenon, shows a widely held practice to be erroneous in someway, enables a promising direction of research for a (broad or narrow) topic, or creates an exciting new technique.

**Paper Topic And Main Contributions:**

This paper aims to build a good reward model (RM) or preference model (PM) that scores how "good" a language model (or human-written) response is.

This is an extremely important problem for the field right now because:
1) Evaluation for long-form answers (like those generated by ChatGPT-style models) is really difficult and coming up with a cheap automatic (model-based) evaluation metric that aligns well with human evaluation is gonna be hugely beneficial for the field.
2) I'm subjectively reluctant to say this but I think objectively, at least in the short term, RLHF has been shown to be a promising direction to "align" language models, and having a good RM/PM is the essential component (this paper doesn't include experiments of RLHF per se, which I think is fine because that'll be too much to ask to pack into a single paper but I'm sure there'll be follow-up papers along this line).

In terms of methodology, this paper first identifies a big flaw in the current paradigm of scoring LLM responses: everything is crammed into a single preference score by human annotators when ranking or evaluating the responses and so it's unclear what's the basis for this score.  The authors then define principles (axioms) that humans desire in longform answers around the concepts of usefulness, relevance, groundedness, truthfulness, and thoroughness. They then construct data to train the PM to learn these axioms in differentiating "better" and "worse" responses.

And I really like the authors' methodology where they follow these axioms to do some nice data work and leverage large-scale data from community QA forums like Reddit and Stack Overflow. Table 1 pretty much summarized how this works. For example, for usefulness, they use upvotes as the proxy to construct training data; for relevance, they construct pairs of on- and off-topic responses (hard negatives retrieved from responses to other questions); for groundedness, the "better" answers are grounded on retrieved contexts while "worse" answers are not, etc. And mining these existing data from online forums also offers good data diversity as compared to just sampling LLM-generated responses with the same underlying LLM.

Training on these constructed data gives good empirical results. The authors evaluated the PM model trained on the full set of data as well as different subsets on various held-out and new test sets. They mainly evaluated ranking performance and how well the PM aligns with human preferences and their 220M PM has shown promising performance. My only concern is whether they did a thorough evaluation on OOD data unseen during training, which hopefully the authors can clarify during the rebuttal.

 Assuming all data and models will be open-sourced, I think this is a great work and will inspire many follow-up studies. And it will be a loss for EMNLP if it doesn't get in :-)

Appendix:
I know the Appendix has many good details which I really should have taken a closer read; but I'm writing this review by squeezing time from my family vacation so I'll admit I only read the main paper. - but I will find time to revisit this paper again after the reviewing deadline when I get back to work.



**Questions For The Authors:**

- I'm curious to see a little more elaboration on where these axioms come from. I know you cited the LaMDA paper, but how did they come up with those principles? "Google AI Principles"??

- Could you remind me what the pos/neg numbers on top of Table 2 represent?

-  What does "(tie)" mean in Table 4?

**Reasons To Accept:**

- Important and timely research problem.
- Very good empirical results.
- The whole paper is well organized and written.
- I think this paper is sending an important message that doing good data work is very valuable in this LLM era - especially since these data are constructed from existing online forums, which is much cheaper than collecting new human preference data.

**Reasons To Reject:**

- I was kind of hoping to see a bit more discussion on the comparison between this work and Anthropic's alignment work (e.g., RLAIF / Constitution AI), where in that case they also found a cheap alternative to actual human preference data. I understand the space constraints so I don't count this as a strong reason to reject, but maybe you could add this in the camera-ready version?

- I probably missed this somewhere, could you help clarify whether the WebGPT comparison / MS MARCO / "Research-Analysis Qs" datasets can be considered OOD? I'm generally interested to know how well can the trained model extrapolate beyond the training data distribution (e.g., those that don't look like coming from online forums). (Also, where does that WebGPT comparison dataset come from? In-house data?) I think it's pretty important for a good PM to be reliable on "all sorts of" distributions (I know it might be difficult to achieve but the evaluation should include results for OOD tests no matter the results are good or bad).

- Consider adding an Ethical Concern section since your work involves human preference studies (you could also add in more details like where do the human annotators come from, how much are they compensated etc.).





**Reproducibility:**

3: Could reproduce the results with some difficulty. The settings of parameters are underspecified or subjectively determined; the training/evaluation data are not widely available.

**Reviewer Confidence:**

4: Quite sure. I tried to check the important points carefully. It's unlikely, though conceivable, that I missed something that should affect my ratings.

---

> ### Author Rebuttal · Authors · 2023-08-29
>
> We would like to thank Reviewer 83C4 for their actionable feedback.
>
> We would like to highlight, as we mentioned to Reviewers k1y7 and sQQz, that we have conducted an additional experiment that compares our PM to two strong baselines:
> 1. LAION  Open-Assistant, a 6.7B-parameter preference model and
> 2. HuggingFace's Stack-Llama reward model that also boasts a 7B-parameter base model.
>
> We report these baselines in an **updated Table 2 in the rebuttal for k1y7 and sQQz**, wherein we found that our PM_0-5 model, with only 220M parameters, outperformed both models even though the 7B-parameter Stack-Llama model from Hugging Face was also trained on Stack Exchange data.
>
> # Reason To Reject 1
>
> > ”a bit more discussion on the comparison between this work and Anthropic's alignment work (e.g., RLAIF / Constitution AI)”:
>
> There is a strong connection between our axiomatic preference models and RLAIF. On the one hand, you can prompt a teacher LLM like GPT-4 to score preferences using a “Constitution” of principles. On the other hand, GPT-4 is high-latency and cost to call for every RLHF training step, so it would be nice to “distill” its preference scores into a faster and cheaper model. We believe our axiomatic approach allows us to **distill a teacher's AI-feedback preference scores into a smaller PM while preserving all the principles in the Constitution**. Interpreted another way, this approach combines learning from human- and AI-generated preference signals in a way that is hopefully better than either source alone.
>
> We have a **preliminary experiment** in this regard: We select a subset of ~70k stackexchange training questions which have chatGPT-generated answers for all the axioms. Then we inference a teacher model’s scores (ChatGPT and GPT-4's) for all the candidate answers using our “critique-then-score" prompt (which in essence is a Constitution since it contained our Principles). We train a T5-base PM_0-5 to learn towards **variable margins** computed from the teacher’s scores aka margin = $log_{10}$(teacher score of pos) - $log_{10}$(teacher score of neg) rather than constant margins. We report these numbers in a **version of Table 3 below** (the numbers in Table 2 did not change much)
>
> |                                    | Ax 2: Open- vs Closed Book | Ax 4: Rel.- vs. Irrel. Context | Ax 5: Combine Top 2 |                |
> |------------------------------------|:--------------------------:|:------------------------------:|:-------------------:|:--------------:|
> |                                    |      Pos > Neg (%)     |        Pos > Neg (%)       |    Comb > Top Ans   | Comb > 2nd Ans |
> | **OpenAsst-RM 6.7B**               |            30.7            |            **92.9**            |         76.1        |      85.1      |
> | **StackLlama RM 7B**               |            38.8            |              47.2              |         57.6        |      66.5      |
> | PM_0 T5-base                       |            70.0            |              30.9              |         25.7        |      34.9      |
> | PM_0-1 T5-base                     |          **77.7**          |              52.8              |         47.0        |      57.7      |
> | PM_0-2 T5-base                     |            76.4            |              82.3              |         66.3        |      80.3      |
> | PM_0-3 T5-base                     |            71.3            |              76.0              |         58.2        |      73.8      |
> | PM_0-4 T5-base                     |            55.1            |              91.4              |         59.7        |      75.8      |
> | PM_0-5 T5-base                     |            53.4            |              92.8              |       **97.4**      |    **98.6**    |
> | **PM_0-5 T5-base Teacher=ChatGPT** |            70.8            |              91.9              |         60.0        |      74.9      |
> | **PM_0-5 T5-base Teacher=GPT4**    |            73.0            |              90.3              |         55.8        |      70.7      |
>
> The initial results are promising: the Teacher-distillation help PM_0-5 on Axiom 2 (where Stack-LLama and Open-Assistant RMs also did very poorly). For Axiom 5, we suspect that the PM_0-5 T5-base may be overfitting to the length of the combined answer. We will complete our investigation by the time of the camera-ready.
>
> # Reason To Reject 2
>
> > ”clarify whether the WebGPT comparison / MS MARCO / "Research-Analysis Qs" datasets can be considered OOD”
>
> Data from Stack Exchange and MS Marco were used for training the PM; even though we test on different segments of the data, these two data sets are considered "in-domain". We do not use data from Reddit/ELI5, WebGPT or Research Analysis questions in training, so these data are considered "out-of-domain". Note that the study focuses on "longform QA" questions and we have not studied how well the trained model would extrapolate to different types of tasks, though it is a subject of future work.
>
> # Reason To Reject 3
> > ”Consider adding an Ethical Concern section”
>
> Yes we will add an Ethics section surrounding the human annotators. The study was approved by our IRB, and the contractor, Scale AI, agreed to adhere to our ethics policies. As part  of that agreement, the annotators were paid at least $15/hr.
>
> # Question 1
> > ”a little more elaboration on where these axioms come from”
>
> In addition to Lamda, we looked at Claude’s Constitutions [1] and Sparrow’s Rules [2] and adapted them to longform QA. In particular, Sparrow has a “stay on topic” rule which we adapted as “relevance”. Groundedness more or less came from the finding [3] that retrieval-augmented search assistants such as BingChat  and Perplexity.ai’s assistant did not properly ground their answers in the retrieved documents.
>
> Axiom 5 “thoroughness” is unique to our work and based on our observation that the top-rated answers on stack exchange often conveyed complementary information (alternative solutions, different viewpoints, etc).
>
> [1] https://www.anthropic.com/index/claudes-constitution
>
> [2] https://storage.googleapis.com/deepmind-media/DeepMind.com/Authors-Notes/sparrow/sparrow-final.pdf Appendix F
>
> [3] Evaluating Verifiability in Generative Search Engines https://arxiv.org/abs/2304.09848
>
> # Question 2
> > ”pos/neg numbers on top of Table 2”
>
>  They represent the average number of positive and negative answers per question, we more clearly denoted this in our updated table. Typically the more negatives, the harder the metric is.
>
> # Question 3
> > ”What does "(tie)" mean in Table 4?”
>
>  In Table 4, (tie) means that GPT-4 gave two answers the same score (which it was allowed to do because scores were integers 1-10). The PMs don’t typically give ties because their numerical precision is fp32.

---

### Official Review · Reviewer_sQQz · 2023-07-30

**Soundness:** 5

**Excitement:**

4: Strong: This paper deepens the understanding of some phenomenon or lowers the barriers to an existing research direction.

**Paper Topic And Main Contributions:**

The paper presents an innovative axiomatic framework designed to generate and augment training pairs, effectively encoding human preferences that may not be present in the original data.  The paper successfully trains a Preference Model (PM) using only 220M parameters. Moreover, the PM demonstrates remarkable performance by effectively scoring both human- and LLM-generated answers, surpassing the capabilities of larger models like GPT-4.

**Reasons To Accept:**

1. The paper's main contribution, the axiomatic framework and the resulting Preference Model, is well-defined and tightly focused on the problem of capturing human preferences more effectively.
2. The perfromance is promising. This paper yields a Preference Model with only about 220M parameters that agrees with gold human-annotated preference labels more often than GPT-4.
3. Large number of experiments are conducted to add credibility to the study.

**Reasons To Reject:**

The contribution is mainly limited into data generation. More interesting methods and exploration of training PM models are expected.

**Reproducibility:**

4: Could mostly reproduce the results, but there may be some variation because of sample variance or minor variations in their interpretation of the protocol or method.

**Reviewer Confidence:**

4: Quite sure. I tried to check the important points carefully. It's unlikely, though conceivable, that I missed something that should affect my ratings.

---

> ### Author Rebuttal · Authors · 2023-08-29
>
> # Reason to Reject 1
> > ”The contribution is mainly limited into data generation”
>
> Thank you for your valuable feedback. The training of a Reward Model (RM) comprises several dimensions:
> 1. Choice of Modeling Objective: Our paper demonstrates that adopting an axiomatic approach representing different dimensions of quality can yield Preference Models (PMs) that are better aligned with human preferences.
> 2. Data: We integrate synthetic data generation with human annotations to create a more comprehensive training dataset. **The data generation process serves as a means to operationalize the concept in (1).**
> 3. Training Algorithm: We acknowledge that this was not the primary focus of our study. However, we introduce a **new experiment** that uses preference scores from GPT-4 as the training target for all axiomatic-generated answers. These “AI feedback” scores allow us to train the PM with a variable (rather than constant) margin, effectively **distilling preference scores from a high-performing teacher model**. We reported those results in the rebuttal to Reviewer 83C4.
>
> We would like to highlight, as we mentioned to Reviewer k1y7, that we have conducted an **additional experiment that compares our model to two strong baselines: (1) LAION  Open-Assistant, a 6.7B-parameter preference model and (2) HuggingFace's Stack-Llama** reward model that also boasts a 7B-parameter base model. Comparing these baselines to our own preference models, we found that our PM_0-5 model, with only 220M parameters, outperformed both models even though the 7B-parameter Stack-Llama model from Hugging Face was also trained on Stack Exchange data. The results are in an updated version of **Table 2 below**:
>
> |                  |      StackX     |          |      r/ELI5     |          |    r/Science    |          |    r/History    |          |     MS Marco    |          |    WebGPT    |
> |------------------|:---------------:|:--------:|:---------------:|:--------:|:---------------:|:--------:|:---------------:|:--------:|:---------------:|:--------:|:------------:|
> |   Avg. Ans per Q | 3.6 pos, 40 neg |          | 4.6 pos, 43 neg |          | 6.5 pos, 42 neg |          | 5.3 pos, 47 neg |          | 1.1 pos, 1k neg |          | 1 pos, 1 neg |
> |           Metric |       MRR       |   NDCG   |       MRR       |   NDCG   |       MRR       |   NDCG   |       MRR       |   NDCG   |       MRR       |   NDCG   |   Accuracy   |
> |      length(Ans) |       15.0      |   35.4   |       6.2       |   27.6   |       7.7       |   30.1   |       15.0      |   37.1   |       n/a       |    n/a   |     56.7     |
> | **OpenAsst-RM 6.7B** |       25.0      |   44.6   |       12.7      |   34.7   |       15.4      |   38.1   |       24.4      |   46.1   |       4.0       |   17.3   |   **76.5**   |
> | **StackLlama RM 7B** |       26.8      |   45.1   |       8.3       |   30.6   |       10.3      |   33.3   |       9.8       |   33.1   |       3.4       |   16.1   |     56.1     |
> | GPT-4 (listwise) |       45.5      |   62.1   |       39.6      |   59.9   |       35.1      |   56.4   |       37.8      |   60.4   |       n/a       |    n/a   |      n/a     |
> |     PM_0 T5-base |       31.2      |   48.6   |       11.1      |   32.6   |       14.8      |   37.0   |       24.0      |   44.5   |       3.9       |   16.9   |     51.1     |
> |   PM_0-1 T5-base |       64.3      |   78.8   |       54.5      |   75.2   |       53.2      |   75.4   |       63.1      |   84.3   |       16.1      |   30.6   |     55.7     |
> |   PM_0-2 T5-base |       65.5      |   79.8   |       55.1      |   76.3   |       51.9      |   74.6   |       61.4      |   83.1   |       9.7       |   25.6   |     57.6     |
> |   PM_0-3 T5-base |       65.3      |   79.5   |       55.0      |   76.0   |       51.4      |   73.9   |       61.1      |   82.8   |       9.4       |   23.7   |     55.4     |
> |   PM_0-4 T5-base |       65.8      |   80.0   |       54.0      |   75.2   |       51.1      |   74.0   |       61.2      |   83.0   |     **25.0**    | **39.3** |     58.6     |
> |   PM_0-5 T5-base |       64.6      |   79.2   |       53.6      |   75.0   |       51.6      |   74.3   |       61.7      |   83.3   |       23.1      |   37.4   |     58.1     |
> |  **PM_0-5 T5-large** |     **66.4**    | **80.8** |     **55.9**    | **77.0** |     **55.4**    | **77.2** |     **64.0**    | **85.4** |       24.3      |   38.9   |     59.1     |
>
> # Errata
> Note, in the submitted paper, we accidentally entered the incorrect scores for PM_0-5 T5-base for StackX and Reddit numbers in Table 2. The corrected numbers are presented in the table above. The changes were small (e.g. Stack X MRR was incorrectly entered as 63.0 and the corrected number is 64.6, r/ELI5 MRR went from 53.4 to 53.6). The model itself did not change, we just misread a row in our spreadsheet.

---

### Meta-Review · Area_Chair_ap5g · 2023-09-19

**Recommendation:** 4

**Metareview:**

This paper investigates methods for guiding reward models to better align with human preference, and develops an axiomatic framework for generating a rich variety of performance signals to uphold them. Reviewers found this paper addressed an important research problem of building reward models that better align with human preferences. All reviewers recognized the strong empirical results with the proposed method that a model trained with 220M parameters agrees with gold human-annotated preference labels more often than GPT-4. Two reviewers recognized the comprehensive experiments conducted by the authors which helped add credibility to the study. Reviewers suggested adding additional comparison with strong baseline models. The authors responded it with additional experiment results against the LAION Open-Assistant model and HuggingFace's Stack-Llama model. The authors should consider adding these results to the revised version of the paper.

---

### Decision · Program_Chairs · 2023-10-07

**Decision:**

Accept-Main

**Comment:**

This paper investigates methods for guiding reward models to better align with human preference, and develops an axiomatic framework for generating a rich variety of performance signals to uphold them. Reviewers found this paper addressed an important research problem of building reward models that better align with human preferences. All reviewers recognized the strong empirical results with the proposed method that a model trained with 220M parameters agrees with gold human-annotated preference labels more often than GPT-4. Two reviewers recognized the comprehensive experiments conducted by the authors which helped add credibility to the study. Reviewers suggested adding additional comparison with strong baseline models. The authors responded it with additional experiment results against the LAION Open-Assistant model and HuggingFace's Stack-Llama model. The authors should consider adding these results to the revised version of the paper.